# Dense-Exponential Random Features: Sharp Positive Estimators of the Gaussian Kernel

**Valerii Likhosherstov**[*†]
University of Cambridge
v.lihosherstov@gmail.com

**Krzysztof Choromanski**[*]
Google DeepMind &
Columbia University
kchoro@google.com

**Avinava Dubey**[*]
Google Research

**Frederick Liu**
Google Research

**Tamas Sarlos**
Google Research

**Adrian Weller**
University of Cambridge &
The Alan Turing Institute

## Abstract

The problem of efficient approximation of a linear operator induced by the Gaussian or softmax kernel is often addressed using *random features* (RFs) which yield an unbiased approximation of the operator's result. Such operators emerge in important applications ranging from kernel methods to efficient Transformers. We propose parameterized, positive, non-trigonometric RFs which approximate Gaussian and softmax-kernels. In contrast to traditional RF approximations, parameters of these new methods can be optimized to reduce the variance of the approximation, and the optimum can be expressed in closed form. We show that our methods lead to variance reduction in practice ($e^{10}$-times smaller variance and beyond) and outperform previous methods in a kernel regression task. Using our proposed mechanism, we also present FAVOR#, a method for self-attention approximation in Transformers. We show that FAVOR# outperforms other random feature methods in speech modelling and natural language processing.

## 1 Introduction

Random feature decomposition is an important technique for the linearization of nonlinear kernel functions with theoretical guarantees such as unbiasedness and concentration around the true kernel value. Linearization allows a significant reduction in computations from quadratic to linear complexity in the size of the operator induced by the kernel. The technique emerged under the name of *random kitchen sinks* (RKS) introduced in [39, 40, 41] and was used in many applications such as kernel SVM [43, 29, 38, 3], dimensionality reduction [18, 1], neural networks [12, 22, 49, 24, 14], function-to-function regression [34], kernel regression [28, 2], nonparametric adaptive control [6], differentially-private ML algorithms [11], operator-valued kernels [32, 8] and semigroup kernels [51]. An in-depth theoretical analysis of random features was performed by [30, 52, 44, 42].

An exciting recent application of random features is in the area of scalable Transformer networks [16, 13, 17, 26], where the self-attention matrix is approximated as a low-rank matrix when the sequence is long. However, the RKS family of methods relies on the Fourier transform, resulting in $\sin$ and $\cos$ types of random features, which were shown to be unsuitable for application in Transformers due to negative values in the low-rank matrix. [16] proposed a solution in the form of positive-valued random features relying on the exponential function (*positive random features, PosRFs*), yielding

---

[*] Equal contribution

[†] Work done while at University of Cambridge. The author is at Waymo now.

37th Conference on Neural Information Processing Systems (NeurIPS 2023).

a method they called *Fast Attention Via Orthogonal positive Random features (FAVOR+)* for self-attention approximation. This solution was improved by [31] by means of a careful choice of the linear combination parameters under the exponent, and the so-called *homogeneity heuristic*, which allows a choice of one set of parameters for all approximated values. The resulting random features were called *generalized exponential random features (GERFs)*, and the corresponding self-attention approximation method was termed *FAVOR++*.

**Contributions:** In this paper, we make a leap forward in the design of positive-valued random features by proposing *dense exponential random features (DERFs)* which contain both PosRFs and GERFs as special cases. Instead of scalar parameters as in GERFs, DERFs rely on matrix parameters and dense quadratic forms inside the exponent. We show how to select parameters of the new random features efficiently without harming the overall subquadratic complexity.

More technically, our contributions are as follows:

1. We show that the homogeneity heuristic of [31] may in fact be viewed not as a heuristic, but a closed-form optimum of the *shifted log-variance objective*.

2. We introduce DERFs and three special instantiations: *asymmetric* DERFs (*ADERFs*), *symmetric* DERFs (*SDERFs*), and *simplified* ADERFs (*SADERFs*). All these instantiations contain GERFs as a special case (Figure 1, left). For each instantiation we prove that a closed-form optimum of the shifted log-variance objective can be found efficiently.

3. We show that our new variants result in lower variance than GERFs and other previous methods in practice (e.g. up to $e^{10}$ times variance improvement as in Figure 1, right). Further, we show that DERFs outperform other random features in kernel regression and Transformer setups (speech modelling and natural language processing). We refer to the DERF-based self-attention approximation method as *FAVOR#*.

## 2 Prerequisites

### 2.1 Scaled softmax kernel and random features

By the *scaled softmax kernel* $K^{(\alpha)} : \mathbb{R}^d \times \mathbb{R}^d \to (0, +\infty)$, where $\alpha \in \mathbb{R}$, we denote a mapping defined as $K^{(\alpha)}(\mathbf{x}, \mathbf{y}) = \exp(\alpha\|\mathbf{x}\|^2 + \mathbf{x}^\top\mathbf{y} + \alpha\|\mathbf{y}\|^2)$ for all $\mathbf{x}, \mathbf{y} \in \mathbb{R}^d$ where $\|\cdot\|$ is an $L_2$-norm. Two important special cases of the scaled softmax kernel are 1) the *Gaussian kernel* $K^{(-1/2)}(\mathbf{x}, \mathbf{y}) = \exp(-\|\mathbf{x} - \mathbf{y}\|^2/2)$ and 2) the *softmax kernel* $K^{(0)}(\mathbf{x}, \mathbf{y}) = \exp(\mathbf{x}^\top\mathbf{y})$. For two sets of vectors $\mathcal{X} = \{\mathbf{x}^{(i)} \in \mathbb{R}^d\}_{i=1}^L$ and $\mathcal{Y} = \{\mathbf{y}^{(j)} \in \mathbb{R}^d\}_{j=1}^L$, by $\mathcal{K}(\mathcal{X}, \mathcal{Y}) \in \mathbb{R}^{L \times L}$ we denote a matrix where $\mathcal{K}^{(\alpha)}(\mathcal{X}, \mathcal{Y})_{i,j} = K^{(\alpha)}(\mathbf{x}^{(i)}, \mathbf{y}^{(j)})$ for all $1 \leq i, j \leq L$.

In this paper, we will be interested in the problem of computing $\mathcal{K}^{(\alpha)}(\mathcal{X}, \mathcal{Y})\mathbf{C}$ where $\mathcal{X}, \mathcal{Y}$ and a matrix $\mathbf{C} \in \mathbb{R}^{L \times n}$ are provided as an input. A naive solution requires $O(L^2(d + n))$ computations for constructing $\mathcal{K}^{(\alpha)}(\mathcal{X}, \mathcal{Y})$ ($O(L^2 d)$) and computing the matrix multiplication $\mathcal{K}^{(\alpha)}(\mathcal{X}, \mathcal{Y}) \times \mathbf{C}$ ($O(L^2 n)$). Instead, we will use an efficient Monte-Carlo approximation of $\mathcal{K}^{(\alpha)}(\mathcal{X}, \mathcal{Y}) \times \mathbf{C}$ using the following notion of *random features (RFs) for the scaled softmax kernel*:

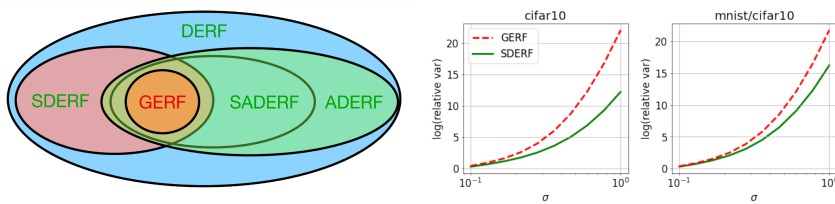

Figure 1: **(left)** Venn diagram of the new types of random features (green) we propose. **(right)** Logarithm of the relative variance of different random feature maps on pairs of vectors sampled from CIFAR10 and MNIST/CIFAR10. A new random feature map SDERF results in a consistent variance reduction of the previous best method GERF, up to $\approx e^{10}$ and $\approx e^5$ times. Figure 2 extends this plot.

**Definition 2.1.** By random features for the scaled softmax kernel $K^{(\alpha)}$, $\alpha \in \mathbb{R}$, we denote a triple $\mathcal{T} = \langle \nu, f^{(1)}, f^{(2)} \rangle$ where $\nu$ is a probability distribution over random objects $\boldsymbol{\omega} \in \Omega$ and $f^{(i)} : \Omega \times \mathbb{R}^d \to \mathbb{R}$, $i \in \{1, 2\}$, are such mappings that, for all $\mathbf{x}, \mathbf{y} \in \mathbb{R}^d$,

$$K^{(\alpha)}(\mathbf{x}, \mathbf{y}) = \mathbb{E}_\nu \left[ f^{(1)}(\boldsymbol{\omega}, \mathbf{x}) f^{(2)}(\boldsymbol{\omega}, \mathbf{y}) \right]. \tag{1}$$

The decomposition of type (1) can be used for an efficient unbiased approximation of $\mathcal{K}^{(\alpha)}(\mathcal{X}, \mathcal{Y})\mathbf{C}$. Let $\boldsymbol{\omega}^{(1)}, \ldots, \boldsymbol{\omega}^{(M)} \in \Omega$ be i.i.d. samples from $\nu$. Define matrices $\mathbf{P}, \mathbf{S} \in \mathbb{R}^{L \times M}$ where for all $1 \le i, j \le L$,

$$\mathbf{P}_{i,:} = M^{-1/2}(f^{(1)}(\boldsymbol{\omega}^{(m)}, \mathbf{x}^{(i)}))_{m=1}^M, \quad \mathbf{S}_{j,:} = M^{-1/2}(f^{(2)}(\boldsymbol{\omega}^{(m)}, \mathbf{y}^{(j)}))_{m=1}^M, \tag{2}$$

where $\mathbf{P}_{i,:}, \mathbf{S}_{j,:} \in \mathbb{R}^d$ are *column vectors* corresponding to the rows of $\mathbf{P}, \mathbf{S}$. Then according to (1), $\widehat{\mathcal{K}} = \mathbf{P}\mathbf{S}^\top$ is an unbiased *Monte Carlo (MC)* approximation of $\mathcal{K}^{(\alpha)}(\mathcal{X}, \mathcal{Y})$ on $M$ samples. The variance $\mathrm{Var}\,\widehat{\mathcal{K}}_{i,j} = M^{-1}\mathrm{Var}_\nu f^{(1)}(\boldsymbol{\omega}, \mathbf{x}^{(i)}) f^{(2)}(\boldsymbol{\omega}, \mathbf{y}^{(j)})$ of this approximation is inversely proportional to $M$, hence $M$ is a tradeoff parameter between computations and precision. Now, $\widehat{\mathcal{K}}\mathbf{C}$ is an unbiased approximation of $\mathcal{K}^{(\alpha)}(\mathcal{X}, \mathcal{Y})\mathbf{C}$ but $\widehat{\mathcal{K}} = \mathbf{P}\mathbf{S}^\top$ is a rank-$M$ matrix, hence computing $\widehat{\mathcal{K}}\mathbf{C}$ has $O(LMn)$ complexity. Assuming that sampling each $\boldsymbol{\omega}^{(m)}$ and computing $f^{(\cdot)}(\cdot, \cdot)$ are $O(d)$ operations, which is usually the case, precomputing $\mathbf{P}$ and $\mathbf{S}$ takes $O(LMd)$ computations, resulting in a total $O(LM(d + n))$ computational complexity. By choosing $M \ll L$, we obtain a significant reduction in computations compared to the exact variant: $O(LM(d + n)) \ll O(L^2(d + n))$.

Operations of type $\mathcal{K}^{(\alpha)}(\mathcal{X}, \mathcal{Y})\mathbf{C}$, especially for the Gaussian kernel $\alpha = -1/2$, emerge in kernel SVM [40], kernel regression [33, 47] and in physics in the form of the Gauss transform [50]. Another important application is in the area of efficient Transformers and is discussed in the next section [16].

## 2.2 Random features for efficient Transformers

RFs found a prominent application in the area of efficient long-sequence Transformers [16]. Transformers rely on a self-attention block for propagating information between elements of the sequence. If the sequence length is $L$ and input matrices are denoted as $\mathbf{Q}, \mathbf{K}, \mathbf{V} \in \mathbb{R}^{L \times d}$ (*queries*, *keys* and *values*), then self-attention outputs the following matrix:

$$\mathbf{Y} = \mathrm{diag}(\mathcal{K}^{(0)}(\mathcal{X}, \mathcal{Y})\mathbf{1}_L)^{-1}\mathcal{K}^{(0)}(\mathcal{X}, \mathcal{Y})\mathbf{V} \in \mathbb{R}^{L \times d}, \tag{3}$$

where $\mathbf{1}_L \in \mathbb{R}^L$ is a vector of all ones, $\mathrm{diag}(\cdot)$ returns a diagonal $(L \times L)$-sized matrix with the argument on the diagonal, $\mathcal{X} = \{\mathbf{x}^{(i)} = d^{-1/4}\mathbf{Q}_{i,:} \in \mathbb{R}^d\}$, and $\mathcal{Y} = \{\mathbf{y}^{(j)} = d^{-1/4}\mathbf{K}_{j,:} \in \mathbb{R}^d\}$. Hence, substitution of $\widehat{\mathcal{K}}$ instead of $\mathcal{K}^{(0)}(\mathcal{X}, \mathcal{Y})$ in (3) reduces computational complexity from $O(L^2 d)$ to $O(LMd)$ ($n = d + 1$). $\mathrm{diag}(\mathcal{K}^{(0)}(\mathcal{X}, \mathcal{Y})\mathbf{1}_L)^{-1}\mathcal{K}^{(0)}(\mathcal{X}, \mathcal{Y})$ is the result of a *softmax* operation performed on rows of $d^{-1/2}\mathbf{Q}\mathbf{K}^\top$.

## 2.3 Existing random features for the softmax kernel

Representation (1) is not unique and different RFs can be proposed for a single $K^{(\alpha)}$. Note that if $\langle \nu, f^{(1)}, f^{(2)} \rangle$ are RFs for $K^{(0)}$, then $\langle \nu, \widehat{f}^{(1)}, \widehat{f}^{(2)} \rangle$ are RFs for $K^{(\alpha)}$ for $\alpha \in \mathbb{R}$ where $\widehat{f}^{(k)}(\boldsymbol{\omega}, \mathbf{x}) = \exp(\alpha\|\mathbf{x}\|^2)f^{(k)}(\boldsymbol{\omega}, \mathbf{x})$. Hereafter we focus on the softmax kernel $K^{(0)}$ without loss of generality.

[15] proposed to use *trigonometric random features (TrigRFs)* from [40] for efficient Transformers:

$$\Omega_{\mathrm{trig}} = \mathbb{R}^{d+1}, \quad \nu_{\mathrm{trig}} = \mathrm{Unif}([0, 2\pi]) \times \mathcal{N}(\mathbf{0}_d, \mathbf{I}_d)^d, \quad f_{\mathrm{trig}}^{(1)}((\theta, \widetilde{\boldsymbol{\omega}}), \mathbf{x}) = \tag{4}$$

$$= \sqrt{2}\exp(\|\mathbf{x}\|^2/2)\cos(\widetilde{\boldsymbol{\omega}}^\top \mathbf{x} + \theta), \quad f_{\mathrm{trig}}^{(2)}((\theta, \widetilde{\boldsymbol{\omega}}), \mathbf{y}) = \sqrt{2}\exp(\|\mathbf{y}\|^2/2)\cos(-\widetilde{\boldsymbol{\omega}}^\top \mathbf{y} + \theta), \tag{5}$$

where $\boldsymbol{\omega} = (\theta, \widetilde{\boldsymbol{\omega}})$, $\mathrm{Unif}(\cdot)$ denotes a uniform distribution on the argument set, $\mathcal{N}(\mathbf{0}_d, \mathbf{I}_d)$ is a multivariate Gaussian distribution with mean $\mathbf{0}_d$ (vector of $d$ zeros) and covariance matrix $\mathbf{I}_d$ (identity matrix of size $d \times d$).

The next iteration of efficient attention approximators [16] observed a problem with TrigRFs (4-5). The *attention matrix* $\mathrm{diag}(\mathcal{K}^{(0)}(\mathcal{X}, \mathcal{Y})\mathbf{1}_L)^{-1}\mathcal{K}^{(0)}(\mathcal{X}, \mathcal{Y})$ from (3) is *right stochastic* meaning that its

entries are nonnegative and each row sums to $1$ due to the normalizing term $\mathrm{diag}(\mathcal{K}^{(0)}(\mathcal{X}, \mathcal{Y})\mathbf{1}_L)^{-1}$. However, since $f_{\mathrm{trig}}^{(\cdot)}$ can be arbitrary real numbers, $\mathbf{P}, \mathbf{S}$ (2) and, therefore, $\widehat{\mathcal{K}}$ can take negative values. Hence, $\mathrm{diag}(\widehat{\mathcal{K}}\mathbf{1}_L)^{-1}\widehat{\mathcal{K}}$ is not right stochastic in general and entries of $\widehat{\mathcal{K}}\mathbf{1}_L$ can take very small and/or negative values resulting in unstable behaviour when inverting $\mathrm{diag}(\widehat{\mathcal{K}}\mathbf{1}_L)^{-1}$. [16] therefore proposed a new type of *positive random features (PosRFs)* which have the form:

$$\Omega_{\mathrm{pos}} = \mathbb{R}^d, \quad \nu_{\mathrm{pos}} = \mathcal{N}(0, 1)^d, \quad f_{\mathrm{pos}}^{(1)}(\boldsymbol{\omega}, \mathbf{x}) = f_{\mathrm{pos}}^{(2)}(\boldsymbol{\omega}, \mathbf{x}) = \exp(\boldsymbol{\omega}^\top \mathbf{x} - \|\mathbf{x}\|^2/2).$$

It is clear that such $f_{\mathrm{pos}}^{(\cdot)}$ only take strictly positive values resulting in the right stochastic $\mathrm{diag}(\widehat{\mathcal{K}}\mathbf{1}_L)^{-1}\widehat{\mathcal{K}}$ and a stable Transformer training procedure.

[31] extended PosRFs, proposing *generalized exponential random features (GERFs)*[1] for $K^{(0)}$:

$$\Omega_{\mathrm{GE}} = \mathbb{R}^d, \ \nu_{\mathrm{GE}} = \mathcal{N}(0, 1)^d, \ f_{\mathrm{GE}}^{(1)}(\boldsymbol{\omega}, \mathbf{x}) = f_{\mathrm{GE}}^{(2)}(\boldsymbol{\omega}, \mathbf{x}) = D \exp(A\|\boldsymbol{\omega}\|^2 + B\boldsymbol{\omega}^\top \mathbf{x} + C\|\mathbf{x}\|^2/2) \quad (6)$$

where $A, B, C, D$ are real numbers[2] satisfying: $1 - 8A > 0$, $B = \sqrt{1 - 4A}$, $C = -0.5$, $D = (1 - 4A)^{d/4}$. [31] express $B, C, D$ through $A$ and find a closed-form equation for the variance of (1):

$$\mathrm{Var}_{\nu_{\mathrm{GE}}} f_{\mathrm{GE}}^{(1)}(\boldsymbol{\omega}, \mathbf{x}) f_{\mathrm{GE}}^{(2)}(\boldsymbol{\omega}, \mathbf{y}) = e^{\mathcal{L}_{\mathrm{GE}}(A, \mathbf{x}, \mathbf{y})} - K^{(0)}(\mathbf{x}, \mathbf{y})^2,$$

$$\mathcal{L}_{\mathrm{GE}}(A, \mathbf{x}, \mathbf{y}) = d \log\left(\frac{1 - 4A}{\sqrt{1 - 8A}}\right) + \frac{2(1 - 4A)}{1 - 8A}\|\mathbf{x} + \mathbf{y}\|^2 - \|\mathbf{x}\|^2 - \|\mathbf{y}\|^2. \quad (7)$$

The minimum variance corresponds to the minimum $\mathcal{L}(A, \mathbf{x}, \mathbf{y})$ since $K^{(0)}(\mathbf{x}, \mathbf{y})^2$ does not depend on $A$. Since $\mathcal{L}(A, \mathbf{x}, \mathbf{y})$ is defined for a single pair of $\mathbf{x}, \mathbf{y}$ and not for sets $\mathcal{X}, \mathcal{Y}$, [31] propose a *homogeneity heuristic* when they replace $\|\mathbf{x} + \mathbf{y}\|^2, \|\mathbf{x}\|^2, \|\mathbf{y}\|^2$ in (7) with averages over $\mathcal{X}, \mathcal{Y}$: $L^{-2}\sum_{i,j}\|\mathbf{x}^{(i)} + \mathbf{y}^{(j)}\|^2, L^{-1}\sum_i\|\mathbf{x}^{(i)}\|^2$ and $L^{-1}\sum_j\|\mathbf{y}^{(j)}\|^2$ respectively. This heuristic is based on the assumption that $\{\mathbf{x}^{(i)}\}$ and $\{\mathbf{y}^{(j)}\}$ are homogeneous and their statistics are tightly concentrated around the mean. After this, the minimum of (7) with respect to $A$ can be found in closed form.

## 3 Dense-exponential random features (DERFs)

We prove that the homogeneity heuristic (Section 2.3) corresponds to a certain minimization problem. Then, we present DERFs which generalize GERFs and provide a tighter solution of that problem.

### 3.1 The objective minimized by GERFs

Our first contribution is showing that the homogeneity heuristic adopted in GERFs is actually an analytic solution of a certain optimization problem. Define

$$\overline{\mathcal{L}}(\boldsymbol{\theta}; \mathcal{X}, \mathcal{Y}, \mathcal{T}) = L^{-2} \sum_{1 \le i, j \le L} \log(\mathrm{Var}_\nu[f^{(1)}(\boldsymbol{\omega}, \mathbf{x}^{(i)}) f^{(2)}(\boldsymbol{\omega}, \mathbf{y}^{(j)})] + K^{(0)}(\mathbf{x}^{(i)}, \mathbf{y}^{(j)})^2), \quad (8)$$

where $\mathcal{T} = \langle \nu, f^{(1)}, f^{(2)} \rangle$ are RFs for the kernel $K^{(0)}$ and $\boldsymbol{\theta}$ are their parameters appearing implicitly in $\nu, f^{(1)}, f^{(2)}$. (8) is a mean log-variance shifted by $K^{(0)}(\mathbf{x}^{(i)}, \mathbf{y}^{(j)})^2$. The best possible value of (8) is $\log K^{(0)}(\mathbf{x}^{(i)}, \mathbf{y}^{(j)})$ which corresponds to all variances $\mathrm{Var} f^{(1)}(\boldsymbol{\omega}, \mathbf{x}^{(i)}) f^{(2)}(\boldsymbol{\omega}, \mathbf{y}^{(j)})$ being zero, meaning that RFs provide exact kernel estimation. Hence, minimization of (8) leads to more precise estimators on $\mathcal{X}, \mathcal{Y}$. We call the loss function $\overline{\mathcal{L}}(\boldsymbol{\theta}; \mathcal{X}, \mathcal{Y}, \mathcal{T})$ the *shifted log-variance* objective. Since $\log$ is concave loss $\overline{\mathcal{L}}$ is conceptually similar to relative standard deviation: the smaller $K^{(0)}(\mathbf{x}^{(i)}, \mathbf{y}^{(j)})$, the higher $\overline{\mathcal{L}}$ is for the same amount of variance $\mathrm{Var} f^{(1)}(\boldsymbol{\omega}, \mathbf{x}^{(i)}) f^{(2)}(\boldsymbol{\omega}, \mathbf{y}^{(j)})$.

If $\mathcal{T}_{\mathrm{GE}} = \langle \nu_{\mathrm{GE}}, f_{\mathrm{GE}}^{(1)}, f_{\mathrm{GE}}^{(2)} \rangle$ are taken in (8), then $\boldsymbol{\theta}_{\mathrm{GE}} = \{A, B, C, D\}$ and $\overline{\mathcal{L}}(\boldsymbol{\theta}_{\mathrm{GE}}; \mathcal{X}, \mathcal{Y}, \mathcal{T}_{\mathrm{GE}}) = L^{-2}\sum_{i,j} \mathcal{L}_{\mathrm{GE}}(A; \mathbf{x}^{(i)}, \mathbf{y}^{(j)})$. Using (7), we get: $\overline{\mathcal{L}}(\boldsymbol{\theta}_{\mathrm{GE}}; \mathcal{X}, \mathcal{Y}, \mathcal{T}_{\mathrm{GE}}) =$

$$d \log\left(\frac{1 - 4A}{\sqrt{1 - 8A}}\right) + \frac{2 - 8A}{1 - 8A} \frac{1}{L^2} \sum_{i,j} \|\mathbf{x}^{(i)} + \mathbf{y}^{(j)}\|^2 - \frac{1}{L} \sum_i \|\mathbf{x}^{(i)}\|^2 - \frac{1}{L} \sum_j \|\mathbf{y}^{(j)}\|^2.$$

---

[1] [31] define these RFs for $K^{(-1/2)}$ but we adapt them for $K^{(0)}$ using the trick mentioned above.

[2] [31] consider a more generalized form when $A, B, C, D$ are complex with an additional parameter $s = \pm 1$, however only the subfamily (6) with $s = 1$ is proposed for use in the Transformer application.

That is, $\mathcal{L}(A; \mathcal{X}, \mathcal{Y})$ coincides with (7) when $\|\mathbf{x} + \mathbf{y}\|^2$, $\|\mathbf{x}\|^2$, $\|\mathbf{y}\|^2$ are replaced by their average statistics computed on $\mathcal{X}, \mathcal{Y}$. Hence, the homogeneity heuristic (Section 2.3) is nothing but minimization of (8). While in general it's unclear how to find a closed-form optimum of $\mathrm{Var}\widehat{\mathcal{K}}$ or $\mathrm{Var}(\widehat{\mathcal{K}}\mathbf{C})$, the global minimum of (8) is feasible and can be computed in $O(1)$ time. Further, in the case of GERF, the values minimizing (8) lead to very good results in large-scale applications of efficient Transformers as shown by [31] without knowing about (8) objective. In the next section we present extensions of GERFs which aim to minimize (8) in closed form.

## 4 Towards DERFs

Dense-exponential random features (DERFs) are an extension of GERFs where scalars $A, B, C$ are replaced with dense matrices. DERFs may be viewed as a generalization that contain the previously introduced classes as special cases. We define DERFs as follows: $\Omega_{\mathrm{DE}} = \mathbb{R}^d$, $\nu_{\mathrm{DE}} = \mathcal{N}(0,1)^d$ and for $k \in \{1, 2\}$:

$$f_{\mathrm{DE}}^{(k)}(\boldsymbol{\omega}, \mathbf{x}) = D \exp(\boldsymbol{\omega}^\top \mathbf{A} \boldsymbol{\omega} + \boldsymbol{\omega}^\top \mathbf{B}^{(k)} \mathbf{x} + \mathbf{x}^\top \mathbf{C}^{(k)} \mathbf{x}),$$

where $\mathbf{B}^{(k)}, \mathbf{C}^{(k)} \in \mathbb{R}^{d \times d}$, $D \in \mathbb{R}$, $\mathbf{A} \in \mathbb{S}_d$ (a set of $d \times d$ real symmetric matrices). Clearly, GERFs with parameters $A, B, C, D$ can be expressed via DERFs with parameters $\mathbf{A} = A\mathbf{I}_d$, $\mathbf{B}^{(1)} = \mathbf{B}^{(2)} = B\mathbf{I}_d$, $\mathbf{C}^{(1)} = \mathbf{C}^{(2)} = C\mathbf{I}_d$, $D$ is unchanged. Our first theoretical result is giving the conditions when $\mathcal{T}_{\mathrm{DE}} = \langle \nu_{\mathrm{DE}}, f_{\mathrm{DE}}^{(1)}, f_{\mathrm{DE}}^{(2)} \rangle$ are valid RFs:

**Theorem 4.1.** *Let the following conditions hold:* $8\mathbf{A} \prec \mathbf{I}_d$, $(\mathbf{B}^{(1)})^\top (\mathbf{I}_d - 4\mathbf{A})^{-1} \mathbf{B}^{(2)} = \mathbf{I}_d$, $\mathbf{C}^{(k)} = -\frac{1}{2}(\mathbf{B}^{(k)})^\top (\mathbf{I}_d - 4\mathbf{A})^{-1} \mathbf{B}^{(k)}$, $D = \det(\mathbf{I}_d - 4\mathbf{A})^{1/4}$ *where* $k \in \{1, 2\}$. *Then* $\mathcal{T}_{\mathrm{DE}}$ *are RFs for* $K^{(0)}$ *and for all* $\mathbf{x}, \mathbf{y} \in \mathbb{R}^d$: $\mathrm{Var}_{\nu_{\mathrm{DE}}} f_{\mathrm{DE}}^{(1)}(\boldsymbol{\omega}, \mathbf{x}) f_{\mathrm{DE}}^{(2)}(\boldsymbol{\omega}, \mathbf{y}) =$

$$D^4 \det(\mathbf{I}_d - 8\mathbf{A})^{-1/2} \exp\left( 2\mathbf{x}^\top \left( \mathbf{C}^{(1)} + (\mathbf{B}^{(1)})^\top (\mathbf{I}_d - 8\mathbf{A})^{-1} \mathbf{B}^{(1)} \right) \mathbf{x} \right.$$

$$\left. + 2\mathbf{y}^\top \left( \mathbf{C}^{(2)} + (\mathbf{B}^{(2)})^\top (\mathbf{I}_d - 8\mathbf{A})^{-1} \mathbf{B}^{(2)} \right) \mathbf{y} + 4\mathbf{x}^\top (\mathbf{B}^{(1)})^\top (\mathbf{I}_d - 8\mathbf{A})^{-1} \mathbf{B}^{(2)} \mathbf{y} \right) - K^{(0)}(\mathbf{x}, \mathbf{y})^2. \quad (9)$$

Our ultimate goal is to find optimal parameters $\mathbf{A}, \mathbf{B}^{(k)}, \mathbf{C}^{(k)}$ and $D$ minimizing the variance of the low-rank approximation of $\mathcal{K}^{(0)}(\mathcal{X}, \mathcal{Y})$ where sets $\mathcal{X}, \mathcal{Y}$ are provided. Our first observation is that we can assume that $\mathbf{A} \in \mathbb{D}_d$ (a set of $d \times d$ real diagonal matrices). Indeed, any symmetric $\mathbf{A}$ can be expressed as $\mathbf{Q}\widetilde{\mathbf{A}}\mathbf{Q}^\top$ where $\mathbf{Q} \in \mathbb{O}_d$ (a set of orthogonal matrices $\{\mathbf{Z} \in \mathbb{R}^{d \times d} \mid \mathbf{Z}^\top \mathbf{Z} = \mathbf{I}_d\}$) and $\widetilde{\mathbf{A}} \in \mathbb{D}_d$. Let $\boldsymbol{\omega} \sim \mathcal{N}(\mathbf{0}_d, \mathbf{I}_d)$. Then, for any $\mathbf{x} \in \mathbb{R}^d$, $k \in \{1, 2\}$,

$$f_{\mathrm{DE}}^{(k)}(\boldsymbol{\omega}, \mathbf{x}) = D \exp(\boldsymbol{\omega}^\top \mathbf{Q}\widetilde{\mathbf{A}}\mathbf{Q}^\top \boldsymbol{\omega} + \boldsymbol{\omega}^\top \mathbf{B}^{(k)} \mathbf{x} + \mathbf{x}^\top \mathbf{C}^{(k)} \mathbf{x}) =$$

$$= D \exp(\widetilde{\boldsymbol{\omega}}^\top \widetilde{\mathbf{A}} \widetilde{\boldsymbol{\omega}} + \widetilde{\boldsymbol{\omega}}^\top \widetilde{\mathbf{B}}^{(k)} \mathbf{x} + \mathbf{x}^\top \mathbf{C}^{(k)} \mathbf{x}) = \widetilde{f}_{\mathrm{DE}}^{(k)}(\widetilde{\boldsymbol{\omega}}, \mathbf{x}),$$

where $\widetilde{\mathbf{B}}^{(k)} = \mathbf{Q}^\top \mathbf{B}^{(k)}$, $\widetilde{\boldsymbol{\omega}} = \mathbf{Q}^\top \boldsymbol{\omega} \sim \mathcal{N}(\mathbf{0}_d, \mathbf{I}_d)$ since the distribution $\boldsymbol{\omega} \sim \mathcal{N}(\mathbf{0}_d, \mathbf{I}_d)$ is *isometric*, i.e. rotation-invariant and $\widetilde{f}_{\mathrm{DE}}^{(k)}$ are DERFs with parameters $\widetilde{\mathbf{A}}, \widetilde{\mathbf{B}}^{(k)}, \mathbf{C}^{(k)}, D$. We conclude that with any $\mathbf{A}$, $f_{\mathrm{DE}}^{(k)}(\boldsymbol{\omega}, \mathbf{x})$ can be expressed as DERFs $\widetilde{f}_{\mathrm{DE}}^{(k)}$ with $\widetilde{\mathbf{A}} \in \mathbb{D}_d$. Hence, hereafter we only consider $\mathbf{A} \in \mathbb{D}_d$ without loss of generality.

Since $\mathbf{B}^{(k)}, \mathbf{C}^{(k)}$ are dense matrices in general, evaluation of $f_{\mathrm{DE}}^{(k)}(\boldsymbol{\omega}, \mathbf{x})$ takes $O(d^2)$ time which is bigger than the $O(d)$ complexity for TrigRFs, PosRFs and GERFs. However, $\mathbf{P}$ and $\mathbf{S}$ matrices (2) can be still computed in a time subquadratic in $L$. For that, precompute $(\mathbf{B}^{(k)})^\top \boldsymbol{\omega}^{(m)}$, $\mathbf{C}^{(1)}\mathbf{x}^{(i)}$, $\mathbf{C}^{(2)}\mathbf{y}^{(j)}$ for all $k \in \{1, 2\}$, $1 \le m \le M$, $1 \le i, j \le L$ in $O((M + L)d^2)$ time. Then, computing $f_{\mathrm{DE}}^{(1)}(\boldsymbol{\omega}^{(m)}, \mathbf{x}^{(i)})$, $f_{\mathrm{DE}}^{(2)}(\boldsymbol{\omega}^{(m)}, \mathbf{y}^{(j)})$ for all $1 \le i, j \le L$, $1 \le m \le M$ takes $O(LMd)$ operations. The complexity of constructing (2) then is $O(L(Md + d^2) + Md^2)$ which is still subquadratic in $L$.

Our goal is to minimize $\overline{\mathcal{L}}(\boldsymbol{\theta}_{\mathrm{DE}}; \mathcal{X}, \mathcal{Y}, \mathcal{T}_{\mathrm{DE}})$ for $\boldsymbol{\theta}_{\mathrm{DE}} = \{\mathbf{A}, \mathbf{B}^{(1)}, \mathbf{B}^{(2)}, \mathbf{C}^{(1)}, \mathbf{C}^{(2)}, D\}$. However, we find that even for a single pair of $\mathbf{x}, \mathbf{y}$ it's unclear how to minimize the variance (9) in closed form. Hence, below we consider special cases where an analytic solution is feasible.

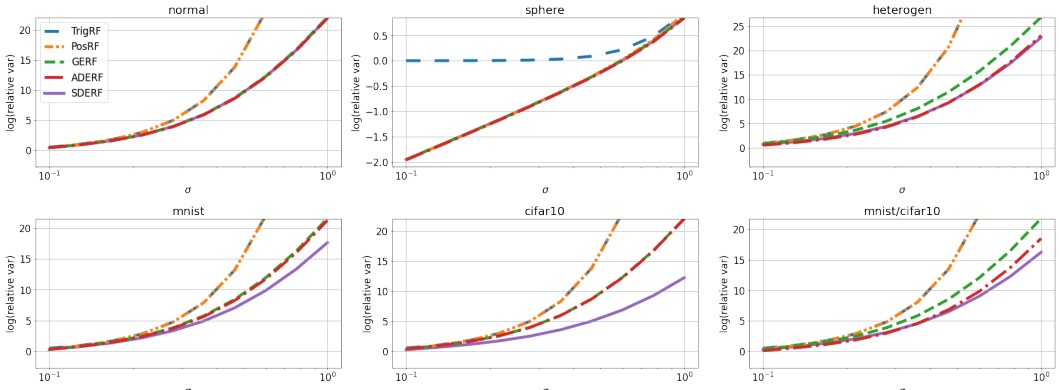

Figure 2: Log of the relative variance of new and existing RF mechanisms, mean value over multiple samples. $0.1 \leq \sigma \leq 1$.

## 4.1 Asymmetric dense-exponential random features

Define RFs $\mathcal{T}_{\text{ADE}} = \langle \nu_{\text{ADE}}, f_{\text{ADE}}^{(1)}, f_{\text{ADE}}^{(2)} \rangle$ in the same way as $\mathcal{T}_{\text{DE}}$ with the only difference that $\mathbf{A} = A\mathbf{I}_d$ where $A \in \mathbb{R}$. We refer to these RFs as *asymmetric dense-exponential RFs (ADERFs)* since $f_{\text{ADE}}^{(1)} \neq f_{\text{ADE}}^{(2)}$ in general. The only additional restriction of ADERFs compared to DERFs is that all diagonal entries of $\mathbf{A} \in \mathbb{D}_d$ are the same. The parameters of $\mathcal{T}_{\text{ADE}}$ are $\boldsymbol{\theta}_{\text{ADE}} = \{A, \mathbf{B}^{(1)}, \mathbf{B}^{(2)}, \mathbf{C}^{(1)}, \mathbf{C}^{(2)}, D\}$. By $\Theta_{\text{ADE}}$ denote a set of all possible $\boldsymbol{\theta}_{\text{ADE}}$'s resulting in correct RFs for the kernel $K^{(0)}$, i.e. satisfying Theorem 4.1 with $\mathbf{A} = A\mathbf{I}_d$. The following result gives an analytic formula for a minimum of $\overline{\mathcal{L}}(\boldsymbol{\theta}_{\text{ADE}}; \mathcal{X}, \mathcal{Y}, \mathcal{T}_{\text{ADE}})$. In the theorem, we use notions of SVD and eigendecomposition of a symmetric matrix [45] (all proofs are in the Appendix).

**Theorem 4.2.** *Let* $\mathcal{X} = \{\mathbf{x}^{(i)} \in \mathbb{R}^d\}_{i=1}^L$, $\mathcal{Y} = \{\mathbf{y}^{(j)} \in \mathbb{R}^d\}_{j=1}^L$. *Let* $\mathbf{M}^{(1)} = \frac{1}{L}\sum_{i=1}^L \mathbf{x}^{(i)}(\mathbf{x}^{(i)})^\top$, $\mathbf{M}^{(2)} = \frac{1}{L}\sum_{j=1}^L \mathbf{y}^{(j)}(\mathbf{y}^{(j)})^\top$. *Suppose that* $\mathbf{M}^{(1)}, \mathbf{M}^{(2)} \in \mathbb{S}_d$ *are nonsingular (implying that* $L \geq d$). *Define* $\mu^{(3)} = d^{-1}L^{-2}\left(\sum_{i=1}^L \mathbf{x}^{(i)}\right)^\top\left(\sum_{j=1}^L \mathbf{y}^{(j)}\right) \in \mathbb{R}$. *For* $k \in \{1, 2\}$, *let* $\mathbf{M}^{(k)} = \mathbf{Q}^{(k)}\boldsymbol{\Lambda}^{(k)}(\mathbf{Q}^{(k)})^\top$ *be eigendecomposition of a symmetric* $\mathbf{M}^{(k)}$ *where* $\mathbf{Q}^{(k)} \in \mathbb{O}_d$. $\boldsymbol{\Lambda}^{(k)} \in \mathbb{D}_d$ *has strictly positive diagonal values since* $\mathbf{M}^{(k)} \succeq 0$ *by definition and* $\mathbf{M}^{(k)}$ *is nonsingular. Let* $\mathbf{U}\boldsymbol{\Sigma}\mathbf{V}^\top$ *be SVD decomposition of* $(\boldsymbol{\Lambda}^{(1)})^{\frac{1}{2}}(\mathbf{Q}^{(1)})^\top\mathbf{Q}^{(2)}(\boldsymbol{\Lambda}^{(2)})^{\frac{1}{2}}$ *where* $\mathbf{U}, \mathbf{V} \in \mathbb{O}^d$, $\boldsymbol{\Sigma} \in \mathbb{D}_d$ *has nonnegative diagonal entries.*

*One of the solutions* $\boldsymbol{\theta}_{\text{ADE}}^* = \{A, \mathbf{B}^{(1)}, \mathbf{B}^{(2)}, \mathbf{C}^{(1)}, \mathbf{C}^{(2)}, D\}$ *of* $\min_{\boldsymbol{\theta}_{\text{ADE}} \in \Theta_{\text{ADE}}} \overline{\mathcal{L}}(\boldsymbol{\theta}_{\text{ADE}}; \mathcal{X}, \mathcal{Y}, \mathcal{T}_{\text{ADE}})$ *is as follows. Set* $\phi = 2d^{-1}\sum_{l=1}^d \boldsymbol{\Sigma}_{l,l} + 2\mu^{(3)}$ *and, for* $k \in \{1, 2\}$,

$$A = \frac{1}{16}\left(1 - 2\phi - \sqrt{(2\phi + 1)^2 + 8\phi}\right), \quad \mathbf{B}^{(1)} = \sqrt{1 - 4A}\boldsymbol{\Sigma}^{1/2}\mathbf{U}^\top(\boldsymbol{\Lambda}^{(1)})^{-1/2}(\mathbf{Q}^{(1)})^\top,$$

$$\mathbf{B}^{(2)} = \sqrt{1 - 4A}\boldsymbol{\Sigma}^{-1/2}\mathbf{U}^\top(\boldsymbol{\Lambda}^{(1)})^{1/2}(\mathbf{Q}^{(1)})^\top, \quad \mathbf{C}^{(k)} = -\frac{1}{2(1 - 4A)}(\mathbf{B}^{(k)})^\top\mathbf{B}^{(k)}, \quad D = (1 - 4A)^{d/4}.$$

*Further, we have:* $\overline{\mathcal{L}}(\boldsymbol{\theta}_{\text{ADE}}^*; \mathcal{X}, \mathcal{Y}, \mathcal{T}_{\text{ADE}}) =$

$$d\left(\log(1 - 4A) - \frac{1}{2}\log(1 - 8A) + 2(1 - 8A)^{-1}\left(d^{-1}\sum_{l=1}^d \boldsymbol{\Sigma}_{l,l} + \mu^{(3)}\right) + 2\mu^{(3)}\right). \tag{10}$$

Theorem 4.2 implies an algorithm for finding $\boldsymbol{\theta}_{\text{ADE}}^*$ efficiently. Namely, compute $\mathbf{M}^{(k)}$, $k \in \{1, 2\}$ ($O(Ld^2)$ time) and $\mu^{(3)}$ ($O(Ld)$ time). Then, perform matrix decompositions to obtain $\mathbf{Q}^{(k)}, \boldsymbol{\Lambda}^{(k)}$, $k \in \{1, 2\}$, and $\mathbf{U}, \boldsymbol{\Sigma}, \mathbf{V}$ in $O(d^3)$ time. After that, $A, \mathbf{B}^{(1)}, \mathbf{B}^{(2)}, \mathbf{C}^{(1)}, \mathbf{C}^{(2)}, D$ can be all evaluated in $O(d^3)$ time using formulae from Theorem 4.2. The total time complexity of the approximation scheme is therefore $O(L(Md + d^2) + Md^2 + d^3)$ which is subquadratic in $L$ as required.

## 4.2 Symmetric dense-exponential random features

Define $\mathcal{T}_{\text{SDE}} = \langle \nu_{\text{SDE}}, f_{\text{SDE}}^{(1)}, f_{\text{SDE}}^{(2)} \rangle$ in the same way as $\mathcal{T}_{\text{DE}}$ with the only difference that $\mathbf{B}^{(1)} = \mathbf{B}^{(2)} = \mathbf{B}$. From the conditions in Theorem 4.1 it follows immediately that also $\mathbf{C}^{(1)} = \mathbf{C}^{(2)} = \mathbf{C}$. Hence, $f_{\text{SDE}}^{(1)} = f_{\text{SDE}}^{(2)}$ and we refer to these RFs as *symmetric dense-exponential RFs (SDERFs)*. The parameters of $\mathcal{T}_{\text{SDE}}$ are $\boldsymbol{\theta}_{\text{SDE}} = \{\mathbf{A}, \mathbf{B}, \mathbf{C}, D\}$. By $\Theta_{\text{SDE}}$ denote a set of all possible $\boldsymbol{\theta}_{\text{SDE}}$'s resulting in correct RFs for the kernel $K^{(0)}$, i.e. satisfying conditions from Theorem 4.1 with $\mathbf{B}^{(k)} = \mathbf{B}$, $\mathbf{C}^{(k)} = \mathbf{C}$, $k \in \{1, 2\}$. The following theorem gives an analytic solution for a global minimum of $\overline{\mathcal{L}}(\boldsymbol{\theta}_{\text{SDE}}; \mathcal{X}, \mathcal{Y}, \mathcal{T}_{\text{SDE}})$.

**Theorem 4.3.** *Let $\mathcal{X} = \{\mathbf{x}^{(i)} \in \mathbb{R}^d\}_{i=1}^L$, $\mathcal{Y} = \{\mathbf{y}^{(j)} \in \mathbb{R}^d\}_{j=1}^L$ and let $\mathbf{M}^{(1)}$, $\mathbf{M}^{(2)}$ be defined as in Theorem 4.2 (but without the restriction of nonsingularity) and define $\boldsymbol{\mu}^{(4)} = \frac{1}{L}\sum_{i=1}^L \mathbf{x}^{(i)} \in \mathbb{R}^d$, $\boldsymbol{\mu}^{(5)} = \frac{1}{L}\sum_{j=1}^L \mathbf{y}^{(j)} \in \mathbb{R}^d$. Further, let $\mathbf{Q}^{(3)}\boldsymbol{\Lambda}^{(3)}(\mathbf{Q}^{(3)})^\top$ be eigendecomposition of a symmetric positive semidefinite matrix $\mathbf{M}^{(1)} + \boldsymbol{\mu}^{(4)}(\boldsymbol{\mu}^{(5)})^\top + \boldsymbol{\mu}^{(5)}(\boldsymbol{\mu}^{(4)})^\top + \mathbf{M}^{(2)}$ where $\mathbf{Q}^{(3)} \in \mathbb{O}_d$ and $\boldsymbol{\Lambda}^{(3)} \in \mathbb{D}_d$ with nonnegative diagonal entries. Let the entries on the diagonal of $\boldsymbol{\Lambda}^{(3)}$ be sorted in the non-ascending order.*

*One of the solutions $\boldsymbol{\theta}_{\text{SDE}}^* = \{\mathbf{A}, \mathbf{B}, \mathbf{C}, D\}$ of $\min_{\boldsymbol{\theta}_{\text{SDE}} \in \Theta_{\text{SDE}}} \overline{\mathcal{L}}(\boldsymbol{\theta}_{\text{SDE}}; \mathcal{X}, \mathcal{Y}, \mathcal{T}_{\text{SDE}})$ is as follows. $\mathbf{A} \in \mathbb{D}_d$, for all $1 \leq l \leq d$: $\mathbf{A}_{l,l} = \frac{1}{16}\left(1 - 2\boldsymbol{\Lambda}_{l,l}^{(3)} - \sqrt{\left(2\boldsymbol{\Lambda}_{l,l}^{(3)} + 1\right)^2 + 8\boldsymbol{\Lambda}_{l,l}^{(3)}}\right)$, $\mathbf{B} = (\mathbf{I}_d - 4\mathbf{A})^{1/2}(\mathbf{Q}^{(3)})^\top$, $\mathbf{C} = -\frac{1}{2}\mathbf{I}_d$, $D = \det(\mathbf{I}_d - 4\mathbf{A})^{1/4}$. Further, we have: $\overline{\mathcal{L}}(\boldsymbol{\theta}_{\text{SDE}}; \mathcal{X}, \mathcal{Y}, \mathcal{T}_{\text{SDE}}) =$*

$$\sum_{l=1}^d \left(\log(1-4\mathbf{A}_{l,l}) - \frac{1}{2}\log(1-8\mathbf{A}_{l,l}) + \left(1 + (1-8\mathbf{A}_{l,l})^{-1}\right)\boldsymbol{\Lambda}_{l,l}^{(3)}\right) - \frac{1}{L}\sum_{i=1}^L \|\mathbf{x}^{(i)}\|^2 - \frac{1}{L}\sum_{j=1}^L \|\mathbf{y}^{(j)}\|^2 \quad (11)$$

Again, Theorem 4.3 implies an algorithm for finding $\boldsymbol{\theta}_{\text{SDE}}^*$ in a time subquadratic in $L$. That is, we can compute $\mathbf{M}^{(1)}, \mathbf{M}^{(2)}, \mu^{(3)}, \boldsymbol{\mu}^{(4)}, \boldsymbol{\mu}^{(5)}$ in $O(Ld^2)$ total time. Then, perform an eigendecomposition to obtain $\mathbf{Q}^{(3)}, \boldsymbol{\Lambda}^{(3)}$ in $O(d^3)$ time. After that, $\mathbf{A}, \mathbf{B}, \mathbf{C}, D$ can be computed in $O(d^3)$ time using formulae from Theorem 4.3. The total time complexity of the approximation scheme is the same as for ADERFs: $O(L(Md + d^2) + Md^2 + d^3)$ or $O(L(Md + d^2) + Md^2)$ if we assume that $L \geq d$.

## 4.3 Simplified ADERFs

While having a compact and closed-form expression, both ADERFs and SDERFs rely on eigende-composition and SVD decompositions: operations for which implementation has not yet matured in popular deep learning libraries with GPU and TPU support. For this reason, we propose *simplified ADERFs (SADERFs)* $\mathcal{T}_{\text{SADE}} = \langle \nu_{\text{SADE}}, f_{\text{SADE}}^{(1)}, f_{\text{SADE}}^{(2)} \rangle$ which extend GERFs but require only basic unary operations. SADERFs are defined via GERFs as follows: $\Omega_{\text{SADE}} = \mathbb{R}^d$, $\nu_{\text{SADE}} = \mathcal{N}(0, 1)^d$, $f_{\text{SADE}}^{(1)}(\boldsymbol{\omega}, \mathbf{x}) = f_{\text{GE}}^{(1)}(\boldsymbol{\omega}, \boldsymbol{\Psi}\mathbf{x})$, $f_{\text{SADE}}^{(2)}(\boldsymbol{\omega}, \mathbf{y}) = f_{\text{GE}}^{(2)}(\boldsymbol{\omega}, \boldsymbol{\Psi}^{-1}\mathbf{y})$ where $\boldsymbol{\Psi} \in \mathbb{D}_d$ is a diagonal matrix with nonzero diagonal entries. First of all, $\mathcal{T}_{\text{SADE}}$ are valid random features for the softmax kernel $K^{(0)}$ since $\mathbb{E}_{\nu_{\text{SADE}}}[f_{\text{SADE}}^{(1)}(\boldsymbol{\omega}, \mathbf{x})f_{\text{SADE}}^{(2)}(\boldsymbol{\omega}, \mathbf{y})] = \mathbb{E}_{\nu_{\text{GE}}}[f_{\text{GE}}^{(1)}(\boldsymbol{\omega}, \boldsymbol{\Psi}\mathbf{x})f_{\text{GE}}^{(2)}(\boldsymbol{\omega}, \boldsymbol{\Psi}^{-1}\mathbf{y})] = K^{(0)}(\boldsymbol{\Psi}\mathbf{x}, \boldsymbol{\Psi}^{-1}\mathbf{y}) = K^{(0)}(\mathbf{x}, \mathbf{y})$, where we use $K^{(0)}(\mathbf{x}, \mathbf{y}) = \exp(\mathbf{x}^\top\mathbf{y})$ by the definition.

We find $\boldsymbol{\Psi}$ by optimizing the objective (8) for $\mathcal{T}_{\text{SADE}}$, the form of which is easily deduced from $\overline{\mathcal{L}}(\boldsymbol{\theta}_{\text{GE}}; \mathcal{X}, \mathcal{Y}, \mathcal{T}_{\text{GE}})$:

$$\overline{\mathcal{L}}(\boldsymbol{\theta}_{\text{SADE}}; \mathcal{X}, \mathcal{Y}, \mathcal{T}_{\text{SADE}}) - d\log\left(\frac{1-4A}{\sqrt{1-8A}}\right) = \frac{2-8A}{1-8A}\frac{1}{L^2}\sum_{i,j}\|\boldsymbol{\Psi}\mathbf{x}^{(i)} + \boldsymbol{\Psi}^{-1}\mathbf{y}^{(j)}\|^2$$

$$-\frac{1}{L}\sum_i(\|\boldsymbol{\Psi}\mathbf{x}^{(i)}\|^2 + \|\boldsymbol{\Psi}^{-1}\mathbf{y}^{(j)}\|^2) = \frac{1}{L^2(1-8A)}\sum_{i,j}\|\boldsymbol{\Psi}\mathbf{x}^{(i)} + \boldsymbol{\Psi}^{-1}\mathbf{y}^{(j)}\|^2 + \frac{2}{L^2}\sum_{i,j}(\mathbf{x}^{(i)})^\top\mathbf{y}^{(j)}, \quad (12)$$

where we move a term not depending on $\boldsymbol{\Psi}$ to the left-hand side. Since $1 - 8A > 0$, we conclude that minimizing (12) is equivalent to minimizing $\sum_{i,j}\|\boldsymbol{\Psi}\mathbf{x}^{(i)} + \boldsymbol{\Psi}^{-1}\mathbf{y}^{(j)}\|^2 =$

$$\sum_l\sum_{i,j}(\boldsymbol{\Psi}_{l,l}\mathbf{x}_l^{(i)} + \boldsymbol{\Psi}_{l,l}^{-1}\mathbf{y}_l^{(j)})^2 = \sum_l\sum_{i,j}(\boldsymbol{\Psi}_{l,l}^2(\mathbf{x}_l^{(i)})^2 + 2\mathbf{x}_l^{(i)}\mathbf{y}_l^{(j)} + \boldsymbol{\Psi}_{l,l}^{-2}(\mathbf{y}_l^{(j)})^2). \quad (13)$$

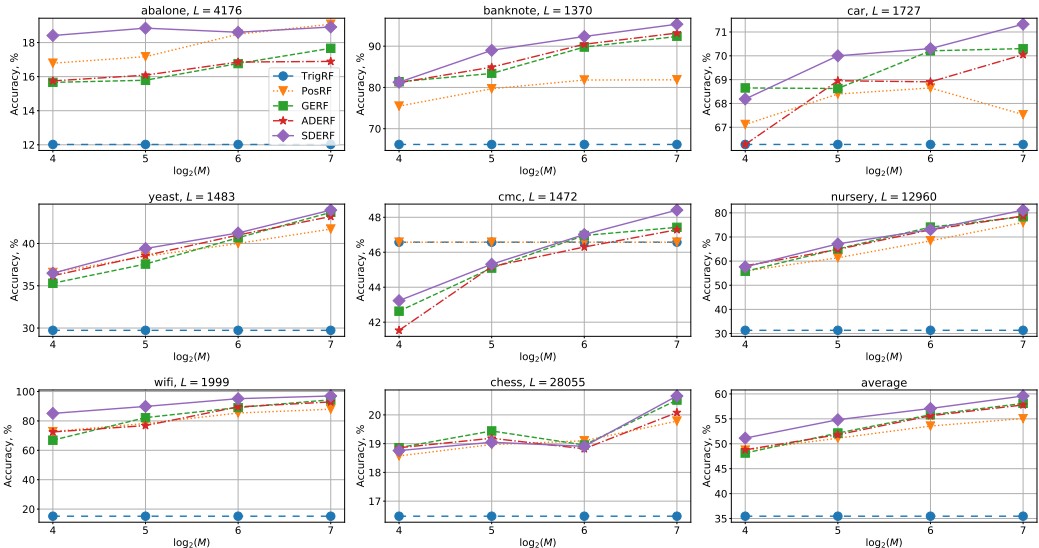

Figure 3: Kernel classification, test accuracy (%). The last plot shows average curves over 8 benchmarks. We observe that our proposed method SDERF, shows the best accuracy across most of the (benchmark, $M$) pairs and also shows the best average performance.

Optimizing (13) reduces to independent optimization problems with respect to $\mathbf{\Psi}_{l,l}$, $1 \leq l \leq d$. Each problem is convex and the solution is found trivially by setting the derivative to zero: for all $1 \leq l \leq d : \mathbf{\Psi}_{l,l}^* = (\sum_j (\mathbf{y}_l^{(j)})^2 / \sum_i (\mathbf{x}_l^{(i)})^2)^{1/4}$. The solution can be computed in $O(dL)$ time, after which the parameters of $f_{\mathrm{GE}}^{(1)}, f_{\mathrm{GE}}^{(2)}$ can be found efficiently as described in Section 2.3.

It is easy to see that $\mathcal{T}_{\mathrm{SADE}}$ are a special case of ADERFs (Section 4.1) which explains their name. Furthermore, the case $\mathbf{\Psi} = \mathbf{I}_d$ reduces $\mathcal{T}_{\mathrm{SADE}}$ to $\mathcal{T}_{\mathrm{GE}}$, hence the latter is a special case of the former. Figure 1 (left) illustrates all the new types of random features in a Venn diagram.

# 5  Experiments

In this section, we evaluate DERFs experimentally in various machine learning applications. More details about each experiment can be found in Appendix B.

## 5.1  Variance comparison

We follow the variance comparison setup from [31]: we sample pairs of vectors $\mathbf{x}, \mathbf{y}$ and compute relative variances of the approximation $\mathrm{Var}\widehat{K}^{(0)}(\mathbf{x}, \mathbf{y})/K^{(0)}(\mathbf{x}, \mathbf{y})$ where $\widehat{K}^{(0)}$ denotes the RF approximation and $\mathrm{Var}\widehat{K}^{(0)}(\mathbf{x}, \mathbf{y})$ is evaluated via (9). We set $d = 64$ as in [31] and take 6 different regimes for sampling $\mathbf{x}, \mathbf{y}$: normal where $\mathbf{x}, \mathbf{y}$ are drawn from $\mathcal{N}(\mathbf{0}_d, \sigma^2 \mathbf{I}_d)$, sphere where $\mathbf{x}, \mathbf{y}$ are drawn uniformly on a sphere $\sigma \mathcal{S}^{d-1}$, heterogen where $\mathbf{x}, \mathbf{y}$ are drawn from different distributions $\mathcal{N}(\mathbf{0}_d, \sigma^2 \mathbf{I}_d)$ and $\mathcal{N}(\sigma \mathbf{1}_d, \sigma^2 \mathbf{I}_d)$. mnist and cifar10 are where $\mathbf{x}, \mathbf{y}$ are random images from MNIST [19] or CIFAR10 [27], resized to $8 \times 8$, scaled by $\sigma > 0$ and flattened. Finally, mnist/cifar10 is a regime where $\mathbf{x}$ is drawn as in mnist and $\mathbf{y}$ is drawn as in cifar10.

We do not report SADERFs since they're a special case of ADERFs (Figure 2). SDERFs outperform or are on par with other methods in all setups – about $e^5$ times better than GERFs in heterogen, mnist and mnist/cifar10 and about $e^{10}$ times better in cifar10. Further, ADERFs outperform GERFs by around $e^3$ times in mnist/cifar10 where $\mathbf{x}$ and $\mathbf{y}$ are drawn "asymmetrically".

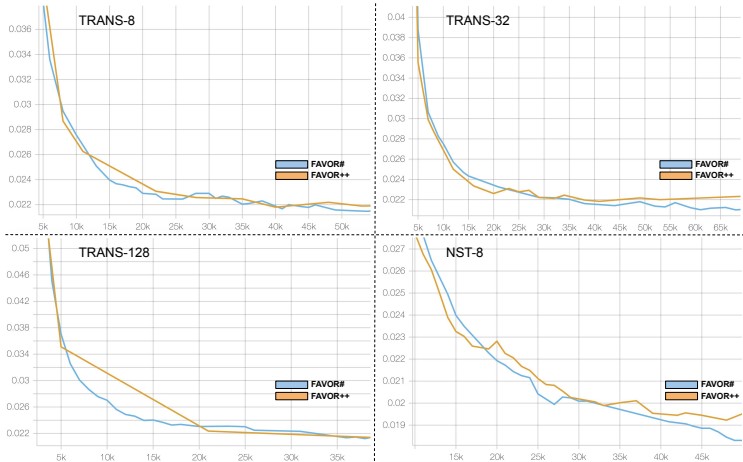

Figure 4: Comparison of FAVOR# using SDRF with FAVOR++ Performer for regular Conformer-Transducer training with $m$ random features (TRANS-$m$) as well as the Noisy Student Training variant with $m$ random features (NST-$m$) on the *LibriSpeech* corpus. We report commonly used normalized word error rate (NWER) metric.

## 5.2 Kernel classification

In this experiment, we compare accuracy of different RF methods in kernel classification on 8 benchmarks from UCI [21], following the setup of [31]. Kernel regression [33, 47] is applied for predicting class probabilities. Training objects are denoted as $\mathbf{u}^{(1)}, \ldots, \mathbf{u}^{(L)} \in \mathbb{R}^d$ and their one-hot labels as $\mathbf{r}^{(1)}, \ldots, \mathbf{r}^{(L)} \in \mathbb{R}^n$. During testing, the goal is to predict the class of a new object $\mathbf{u}^*$ as $\operatorname{argmax}_{1 \le l \le n} \mathbf{r}^*$ where $\mathbf{r}^* = \sum_{i=1}^{L} K^{(-0.5)}(\sigma \mathbf{u}^*, \sigma \mathbf{u}^{(i)}) \mathbf{r}^{(i)}$ and $\sigma > 0$ is tuned on the validation set. With $O(nLM)$ preprocessing, RFs are used to find an unbiased approximation of $\mathbf{r}^*$ in $O(nM)$ instead of $O(nL)$ exact computation. For each benchmark, we range the values of $M$ from $2^4$ to $2^7$ (Figure 3). We observe that SDERF, which is proposed in this paper, shows the best accuracy across most of the (benchmark, $M$) pairs and also shows the best average performance.

## 5.3 DERFs for long-sequence Transformers

In this section, we evaluate DERFs for self-attention approximation in Performer-Transformer training setups [16]. We refer to the DERF-based self-attention approximation method as *FAVOR#*.

### 5.3.1 Speech modelling

In our first set of experiments, we focus on speech models. We train Performer-encoders and test them on the *LibriSpeech* corpus [35], commonly used for benchmarking speech models. We considered two Transformers architectures/training setups: **(a)** Conformer-Transducer [23] trained in a regular way (TRANS) as well as: **(b)** the Noisy Student Training (NST) variant introduced in [36]. We compare "performized" variants of these architectures, applying FAVOR# with SDERF (since it worked best in the previous setups) as well as FAVOR++ [31].

In the first setting, we see that FAVOR# consistently outperforms FAVOR++ for smaller $m$ (where reduced variance of the softmax-kernel estimation is more critical) and both achieve similar scores for larger $m$. In the NST-experiment, we focused on the smaller $m$ variant, where FAVOR# again beats FAVOR++. All results are presented in Fig. 4.

### 5.3.2 Natural language processing

The General Language Understanding Evaluation (GLUE) benchmark [46] consists of 8 different natural language understanding tasks with the sequence length ranging from 32 to 128. We use this to test the performance of different low rank attention methods on NLP tasks. We used the same training parameters as mentioned in [20] (see Appendix B.4.2 for details). We warm start all low-rank

Table 1: GLUE Dev results on base sized models. Number of training examples is reported below each task. MCC score is reported for CoLA, F1 score is reported for MRPC, Spearman correlation is reported for STS-B, and accuracy scores are reported for the other tasks. The **best** result, second best.

| System | MNLI(m) 392k | QQP 363k | QNLI 108k | SST-2 67k | CoLA 8.5k | STS-B 5.7k | MRPC 3.5k | RTE 2.5k |
|---|---|---|---|---|---|---|---|---|
| Local Attention [37] | 75.68 | 86.52 | 87.31 | 89.71 | 61.07 | 83.65 | 75.47 | 59.84 |
| Global + Sliding Wndw [4, 53] | 81.82 | 90.35 | 89.91 | 92.43 | **61.81** | 85.58 | 87.80 | 67.15 |
| ELU [26] | 82.58 | 90.05 | 89.81 | 92.43 | 58.63 | 87.91 | 87.50 | 67.15 |
| RELU [16] | 82.49 | **90.71** | 89.68 | 92.32 | 57.57 | **88.15** | 87.25 | 68.95 |
| FAVOR+ [16] | 77.69 | 86.69 | 89.41 | 91.80 | 54.87 | 83.78 | 80.73 | 66.19 |
| FAVOR++ [31] | 82.29 | 90.43 | 89.73 | 92.20 | 58.85 | 85.90 | **88.73** | 67.63 |
| FAVOR# | **82.69** | 90.68 | **90.01** | **92.53** | 59.33 | 85.48 | 87.99 | **69.68** |

Transformers with a pre-trained BERT-base model checkpoint [20], thus contrasting how well the low rank methods approximate the softmax kernel.

We compared FAVOR++ [31], FAVOR+ [16], ELU [26], ReLU [16] variants of the Performers [16] and sparse attention [37, 4] against the FAVOR# variant and report the results in Table 1. We couldn't use SDERF in this setup because eigendecomposition led to errors on TPUs due to a different implementation compared to the speech modelling experiment. For this reason, we used SADERF which doesn't require any matrix decompositions. On most tasks we find that FAVOR# is the best or second best performing variant showcasing its effectiveness in modelling the softmax kernel for Transformers.

## 6    Conclusion

We proposed an extension of generalized exponential random features (GERFs) for the Gaussian and softmax kernels: dense-exponential random features (DERFs). DERFs employ matrix parameters and are more flexible than GERFs. We evaluated DERFs in several applications such as kernel regression and two Transformers training setups, demonstrating downstream performance benefits.

**Limitations & broader impact.** Optimizing matrix parameters in the most general formulation of DERFs (Theorem 4.1) could lead to further variance reductions. It remains an open question how to find a closed form solution of the optimum which we leave to future work. Our work is primarily theoretical but has broad applications. A prominent application is efficient Transformers which should be used responsibly due to their potential for misuse and significant societal impact, and carbon footprint [48, 5, 10].

## 7    Acknowledgements

V. L. acknowledges support from the Cambridge Trust and DeepMind. V. L. was part-time employed by Google while a PhD student. A.W. acknowledges support from a Turing AI Fellowship under EPSRC grant EP/V025279/1, The Alan Turing Institute, and the Leverhulme Trust via CFI.

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

# A Proofs

## A.1 Proof of Theorem 4.1

*Proof.* By the definition of $\langle \nu_{\mathrm{DE}}, f_{\mathrm{DE}}^{(1)}, f_{\mathrm{DE}}^{(2)} \rangle$, we have:

$$
\mathbb{E}_{\nu_{\mathrm{DE}}} f_{\mathrm{DE}}^{(1)}(\boldsymbol{\omega}, \mathbf{x}) f_{\mathrm{DE}}^{(2)}(\boldsymbol{\omega}, \mathbf{y}) = (2\pi)^{-\frac{d}{2}} D^2 \int_{\mathbb{R}^d} \exp\left(-\frac{1}{2}\|\boldsymbol{\omega}\|^2 + 2\boldsymbol{\omega}^\top \mathbf{A}\boldsymbol{\omega}\right.
$$

$$
+ \boldsymbol{\omega}^\top (\mathbf{B}^{(1)}\mathbf{x} + \mathbf{B}^{(2)}\mathbf{y}) + \mathbf{x}^\top \mathbf{C}^{(1)}\mathbf{x} + \mathbf{y}^\top \mathbf{C}^{(2)}\mathbf{y}\bigg) d\boldsymbol{\omega}
$$

$$
= (2\pi)^{-\frac{d}{2}} D^2 \exp\left(\mathbf{x}^\top \mathbf{C}^{(1)}\mathbf{x} + \mathbf{y}^\top \mathbf{C}^{(2)}\mathbf{y}\right)
$$

$$
\times \int_{\mathbb{R}^d} \exp\left(-\frac{1}{2}\boldsymbol{\omega}^\top (\mathbf{I}_d - 4\mathbf{A})\boldsymbol{\omega} + \boldsymbol{\omega}^\top (\mathbf{B}^{(1)}\mathbf{x} + \mathbf{B}^{(2)}\mathbf{y})\right) d\boldsymbol{\omega}.
$$

Since $8\mathbf{A} \prec \mathbf{I}_d$, we have $4\mathbf{A} \prec 0.5\mathbf{I}_d \prec \mathbf{I}_d$, meaning that $\mathbf{I}_d - 4\mathbf{A}$ is positive definite and invertible. The following identity is straightforward to check:

$$
-\frac{1}{2}\boldsymbol{\omega}^\top (\mathbf{I}_d - 4\mathbf{A})\boldsymbol{\omega} + \boldsymbol{\omega}^\top (\mathbf{B}^{(1)}\mathbf{x} + \mathbf{B}^{(2)}\mathbf{y}) = -\frac{1}{2}(\boldsymbol{\omega} - \boldsymbol{\mu})^\top \boldsymbol{\Sigma}^{-1}(\boldsymbol{\omega} - \boldsymbol{\mu}) + \frac{1}{2}\boldsymbol{\mu}^\top \boldsymbol{\Sigma}^{-1}\boldsymbol{\mu},
$$

$$
\boldsymbol{\Sigma} = (\mathbf{I}_d - 4\mathbf{A})^{-1}, \quad \boldsymbol{\mu} = \boldsymbol{\Sigma}(\mathbf{B}^{(1)}\mathbf{x} + \mathbf{B}^{(2)}\mathbf{y}).
$$

Therefore, we have:

$$
\mathbb{E}_{\nu_{\mathrm{DE}}} f_{\mathrm{DE}}^{(1)}(\boldsymbol{\omega}, \mathbf{x}) f_{\mathrm{DE}}^{(2)}(\boldsymbol{\omega}, \mathbf{y}) = (2\pi)^{-d/2} D^2 \exp\left(\mathbf{x}^\top \mathbf{C}^{(1)}\mathbf{x} + \mathbf{y}^\top \mathbf{C}^{(2)}\mathbf{y} + \frac{1}{2}\boldsymbol{\mu}^\top \boldsymbol{\Sigma}^{-1}\boldsymbol{\mu}\right)
$$

$$
\times \int_{\mathbb{R}^d} \exp\left(-\frac{1}{2}(\boldsymbol{\omega} - \boldsymbol{\mu})^\top \boldsymbol{\Sigma}^{-1}(\boldsymbol{\omega} - \boldsymbol{\mu})\right) d\boldsymbol{\omega}.
$$

Next, we use the fact that the integral of the multivariate Gaussian distribution with mean $\boldsymbol{\mu}$ and variance $\boldsymbol{\Sigma}$ is 1:

$$
(2\pi)^{-d/2} \det(\boldsymbol{\Sigma})^{-1/2} \int_{\mathbb{R}^d} \exp\left(-\frac{1}{2}(\boldsymbol{\omega} - \boldsymbol{\mu})^\top \boldsymbol{\Sigma}^{-1}(\boldsymbol{\omega} - \boldsymbol{\mu})\right) d\boldsymbol{\omega} = 1.
$$

From that we conclude:

$$
\mathbb{E}_{\nu_{\mathrm{DE}}} f_{\mathrm{DE}}^{(1)}(\boldsymbol{\omega}, \mathbf{x}) f_{\mathrm{DE}}^{(2)}(\boldsymbol{\omega}, \mathbf{y}) = D^2 \det(\boldsymbol{\Sigma})^{1/2} \exp\left(\mathbf{x}^\top \mathbf{C}^{(1)}\mathbf{x} + \mathbf{y}^\top \mathbf{C}^{(2)}\mathbf{y} + \frac{1}{2}\boldsymbol{\mu}^\top \boldsymbol{\Sigma}^{-1}\boldsymbol{\mu}\right)
$$

$$
= D^2 \det(\mathbf{I}_d - 4\mathbf{A})^{-1/2} \exp\left(\mathbf{x}^\top \mathbf{C}^{(1)}\mathbf{x} + \mathbf{y}^\top \mathbf{C}^{(2)}\mathbf{y}\right.
$$

$$
+ \frac{1}{2}(\mathbf{B}^{(1)}\mathbf{x} + \mathbf{B}^{(2)}\mathbf{y})^\top (\mathbf{I}_d - 4\mathbf{A})^{-1}(\mathbf{B}^{(1)}\mathbf{x} + \mathbf{B}^{(2)}\mathbf{y})\bigg)
$$

$$
= D^2 \det(\mathbf{I}_d - 4\mathbf{A})^{-1/2} \exp\left(\mathbf{x}^\top \left(\mathbf{C}^{(1)} + \frac{1}{2}(\mathbf{B}^{(1)})^\top (\mathbf{I}_d - 4\mathbf{A})^{-1}\mathbf{B}^{(1)}\right)\mathbf{x}\right.
$$

$$
+ \mathbf{y}^\top \left(\mathbf{C}^{(2)} + \frac{1}{2}(\mathbf{B}^{(2)})^\top (\mathbf{I}_d - 4\mathbf{A})^{-1}\mathbf{B}^{(2)}\right)\mathbf{y} + \mathbf{x}^\top (\mathbf{B}^{(1)})^\top (\mathbf{I}_d - 4\mathbf{A})^{-1}\mathbf{B}^{(2)}\mathbf{y}\bigg).
$$

Based on this expression, we conclude that, indeed, $\mathbb{E}_{\nu_{\mathrm{DE}}} f_{\mathrm{DE}}^{(1)}(\boldsymbol{\omega}, \mathbf{x}) f_{\mathrm{DE}}^{(2)}(\boldsymbol{\omega}, \mathbf{y}) = K^{(0)}(\mathbf{x}, \mathbf{y})$ for all $\mathbf{x}, \mathbf{y} \in \mathbb{R}^d$ if the conditions from theorem's statement are satisfied.

Next, we calculate expression for the variance. For any random variable $Z$, $\mathrm{Var}\, Z = \mathbb{E}Z^2 - (\mathbb{E}Z)^2$. In particular, if $Z = f_{\mathrm{DE}}^{(1)}(\boldsymbol{\omega}, \mathbf{x}) f_{\mathrm{DE}}^{(2)}(\boldsymbol{\omega}, \mathbf{y})$, $\boldsymbol{\omega} \sim \nu_{\mathrm{DE}}$, we get:

$$
\mathrm{Var}_{\nu_{\mathrm{DE}}} f_{\mathrm{DE}}^{(1)}(\boldsymbol{\omega}, \mathbf{x}) f_{\mathrm{DE}}^{(2)}(\boldsymbol{\omega}, \mathbf{y}) = \mathbb{E}_{\nu_{\mathrm{DE}}} f_{\mathrm{DE}}^{(1)}(\boldsymbol{\omega}, \mathbf{x})^2 f_{\mathrm{DE}}^{(2)}(\boldsymbol{\omega}, \mathbf{y})^2 - \left(\mathbb{E}_{\nu_{\mathrm{DE}}} f_{\mathrm{DE}}^{(1)}(\boldsymbol{\omega}, \mathbf{x}) f_{\mathrm{DE}}^{(2)}(\boldsymbol{\omega}, \mathbf{y})\right)^2
$$

$$
= \mathbb{E}_{\nu_{\mathrm{DE}}} f_{\mathrm{DE}}^{(1)}(\boldsymbol{\omega}, \mathbf{x})^2 f_{\mathrm{DE}}^{(2)}(\boldsymbol{\omega}, \mathbf{y})^2 - K^{(0)}(\mathbf{x}, \mathbf{y})^2.
$$

We have:

$$\mathbb{E}_{\nu_{\mathrm{DE}}} f_{\mathrm{DE}}^{(1)}(\boldsymbol{\omega}, \mathbf{x})^2 f_{\mathrm{DE}}^{(2)}(\boldsymbol{\omega}, \mathbf{y})^2 = (2\pi)^{\frac{d}{2}} D^4 \int_{\mathbb{R}^d} \exp\Big(-\frac{1}{2}\|\boldsymbol{\omega}\|^2 + 4\boldsymbol{\omega}^\top \mathbf{A}\boldsymbol{\omega} + 2\boldsymbol{\omega}^\top (\mathbf{B}^{(1)}\mathbf{x} + \mathbf{B}^{(2)}\mathbf{y})$$

$$+ 2\mathbf{x}^\top \mathbf{C}^{(1)}\mathbf{x} + 2\mathbf{y}^\top \mathbf{C}^{(2)}\mathbf{y}\Big) d\boldsymbol{\omega} = (2\pi)^{\frac{d}{2}} D^4 \exp\Big(2\mathbf{x}^\top \mathbf{C}^{(1)}\mathbf{x} + 2\mathbf{y}^\top \mathbf{C}^{(2)}\mathbf{y}\Big)$$

$$\times \int_{\mathbb{R}^d} \exp\Big(-\frac{1}{2}\boldsymbol{\omega}^\top (\mathbf{I}_d - 8\mathbf{A})\boldsymbol{\omega} + 2\boldsymbol{\omega}^\top (\mathbf{B}^{(1)}\mathbf{x} + \mathbf{B}^{(2)}\mathbf{y})\Big) d\boldsymbol{\omega}.$$

Evaluation of the integral above can be done in the same way as calculation of $\mathbb{E}_{\nu_{\mathrm{DE}}} f_{\mathrm{DE}}^{(1)}(\boldsymbol{\omega}, \mathbf{x}) f_{\mathrm{DE}}^{(2)}(\boldsymbol{\omega}, \mathbf{y})$, noticing that $\mathbf{I}_d - 8\mathbf{A}$ is positive definite and invertible. The result is as follows:

$$\mathbb{E}_{\nu_{\mathrm{DE}}} f_{\mathrm{DE}}^{(1)}(\boldsymbol{\omega}, \mathbf{x})^2 f_{\mathrm{DE}}^{(2)}(\boldsymbol{\omega}, \mathbf{y})^2 = D^4 \det(\mathbf{I}_d - 8\mathbf{A})^{-1/2}$$

$$\times \exp\Big(2\mathbf{x}^\top \Big(\mathbf{C}^{(1)} + (\mathbf{B}^{(1)})^\top (\mathbf{I}_d - 8\mathbf{A})^{-1}\mathbf{B}^{(1)}\Big)\mathbf{x}$$

$$+ 2\mathbf{y}^\top \Big(\mathbf{C}^{(2)} + (\mathbf{B}^{(2)})^\top (\mathbf{I}_d - 8\mathbf{A})^{-1}\mathbf{B}^{(2)}\Big)\mathbf{y} + 4\mathbf{x}^\top (\mathbf{B}^{(1)})^\top (\mathbf{I}_d - 8\mathbf{A})^{-1}\mathbf{B}^{(2)}\mathbf{y}\Big).$$

We conclude that the variance expression given in the theorem's statement is correct. □

## A.2 Important lemma

Below, we prove an important lemma which is used in the subsequent proofs:

**Lemma A.1.** *Consider a function* $f : (-\infty, \frac{1}{8})$ *defined as*

$$f(A) = \log(1 - 4A) - \frac{1}{2}\log(1 - 8A) + \frac{\phi}{1 - 8A} \tag{14}$$

*where* $\phi \geq 0$. *Then, the minimum of* $f$ *on* $(-\infty, \frac{1}{8})$ *is achieved at*

$$A^* = \frac{1}{16}\left(1 - 2\phi - \sqrt{(2\phi + 1)^2 + 8\phi}\right). \tag{15}$$

*Proof.* Set $\gamma = (1 - 8A)^{-1} \in (0, +\infty)$. Note that there is a one-to-one correspondence between $\gamma \in (0, +\infty)$ and $A \in (-\infty, \frac{1}{8})$. Hence, we can substitute $\gamma^{-1} = 1 - 8A$ and $1 - 4A = ((1 - 8A) + 1)/2 = (\gamma^{-1} + 1)/2 = \frac{1+\gamma}{2\gamma}$ in (14) and equivalently perform minimization with respect to $\gamma$:

$$\min_{\gamma \in (0, +\infty)} h(\gamma) = \log\left(\frac{\gamma + 1}{2\gamma}\right) + \frac{1}{2}\log\gamma + \phi\gamma = \log(\gamma + 1) - \frac{1}{2}\log\gamma - \log 2 + \phi\gamma.$$

For $h(\cdot)$'s derivative, we have:

$$h'(\gamma) = \phi + \frac{1}{\gamma + 1} - \frac{1}{2\gamma} = \phi + \frac{\gamma - 1}{2\gamma(\gamma + 1)} \tag{16}$$

$$= \frac{2\phi\gamma(\gamma + 1) + \gamma - 1}{2\gamma(\gamma + 1)} = \frac{2\phi\gamma^2 + (2\phi + 1)\gamma - 1}{2\gamma(\gamma + 1)}. \tag{17}$$

Based on (16), we see that $h'(\gamma) \to -\infty$ as $\gamma \to 0$ and $h'(\gamma) > \phi \geq 0$ for all $\gamma > 1$. Hence, we conclude that $h(\cdot)$ is bounded from below on $(0, +\infty)$ and the global minimum $\gamma^*$ on $(0, +\infty)$ exists and it is one of the points satisfying $h'(\gamma^*) = 0$. Hence, it's one of the positive roots of the polynomial in numerator of (17).

If $\phi = 0$, there is a single root $\gamma^* = 1$ of the polynomial in the numerator of (17), hence it is a global minimum of $h(\cdot)$. If $\phi > 0$, then there are two roots of the polynomial in the numerator of (17):

$$\gamma_-^* = \frac{-(2\phi + 1) - \sqrt{(2\phi + 1)^2 + 8\phi}}{4\phi},$$

$$\gamma_+^* = \frac{-(2\phi+1) + \sqrt{(2\phi+1)^2 + 8\phi}}{4\phi}. \tag{18}$$

Note that, if $\phi > 0$, then $2\phi + 1 > 0$ and $(2\phi + 1)^2 + 8\phi \geq (2\phi + 1)^2$. Hence, $\gamma_-^* < 0$ and $\gamma_+^* > 0$. We conclude that $\gamma^* = \gamma_+^*$ is the minimum of $h(\cdot)$ on $(0, +\infty)$. We multiply numerator and denominator of (18)'s right hand side by $(2\phi + 1) + \sqrt{(2\phi + 1)^2 + 8\phi} > 0$:

$$\gamma^* = \gamma_+^* = \frac{((2\phi+1)^2 + 8\phi) - (2\phi+1)^2}{4\phi\left((2\phi+1) + \sqrt{(2\phi+1)^2 + 8\phi}\right)} = \frac{2}{2\phi + 1 + \sqrt{(2\phi+1)^2 + 8\phi}}. \tag{19}$$

Note that the right hand side of (19) is equivalent to (18) when $\phi > 0$ but also holds for the case when $\phi = 0$ (i.e. when $\gamma^* = 1$). We conclude that $f(\cdot)$ is minimized at $\mathbf{A}^* = \frac{1}{8}(1 - (\gamma^*)^{-1})$ since $\gamma^* = (1 - 8A^*)^{-1}$. It's easy to see that (15) follows from (19) directly. $\qquad\square$

### A.3 Proof of Theorem 4.2

*Proof.* With $\mathbf{A} = A\mathbf{I}_d$, the conditions from Theorem 4.1 read as

$$8A < 1, \quad \frac{1}{1-4A}(\mathbf{B}^{(1)})^\top \mathbf{B}^{(2)} = \mathbf{I}_d, \quad \mathbf{C}^{(k)} = -\frac{1}{2(1-4A)}(\mathbf{B}^{(k)})^\top \mathbf{B}^{(k)}, \ D = (1-4A)^{d/4} \tag{20}$$

for $k \in \{1, 2\}$. And the variance expression (9) for all $\mathbf{x}, \mathbf{y} \in \mathbb{R}^d$ transforms into

$$\mathrm{Var}_{\nu_{\mathrm{ADE}}} f_{\mathrm{ADE}}^{(1)}(\boldsymbol{\omega}, \mathbf{x}) f_{\mathrm{ADE}}^{(2)}(\boldsymbol{\omega}, \mathbf{y}) = D^4(1-8A)^{-d/2} \exp\left(2\mathbf{x}^\top\left(\mathbf{C}^{(1)} + \frac{1}{1-8A}(\mathbf{B}^{(1)})^\top \mathbf{B}^{(1)}\right)\mathbf{x}\right.$$

$$\left. + 2\mathbf{y}^\top\left(\mathbf{C}^{(2)} + \frac{1}{1-8A}(\mathbf{B}^{(2)})^\top \mathbf{B}^{(2)}\right)\mathbf{y} + \frac{4}{1-8A}\mathbf{x}^\top(\mathbf{B}^{(1)})^\top \mathbf{B}^{(2)}\mathbf{y}\right) - K^{(0)}(\mathbf{x}, \mathbf{y})^2.$$

We express $\mathbf{C}^{(k)}$ through $A, \mathbf{B}^{(k)}$ and $D$ through $A$ using (20) in the equation above:

$$\mathrm{Var}_{\nu_{\mathrm{ADE}}} f_{\mathrm{ADE}}^{(1)}(\boldsymbol{\omega}, \mathbf{x}) f_{\mathrm{ADE}}^{(2)}(\boldsymbol{\omega}, \mathbf{y}) = \left(\frac{1-4A}{\sqrt{1-8A}}\right)^d \exp\left(\left(\frac{2}{1-8A} - \frac{1}{1-4A}\right)\mathbf{x}^\top(\mathbf{B}^{(1)})^\top \mathbf{B}^{(1)}\mathbf{x}\right.$$

$$\left. + \left(\frac{2}{1-8A} - \frac{1}{1-4A}\right)\mathbf{y}^\top(\mathbf{B}^{(2)})^\top \mathbf{B}^{(2)}\mathbf{y} + \frac{4}{1-8A}\mathbf{x}^\top(\mathbf{B}^{(1)})^\top \mathbf{B}^{(2)}\mathbf{y}\right) - K^{(0)}(\mathbf{x}, \mathbf{y})^2.$$

Since $\frac{1}{1-4A}(\mathbf{B}^{(1)})^\top \mathbf{B}^{(2)}$ is a full-rank matrix $\mathbf{I}_d$ (20), both $\mathbf{B}^{(1)}$ and $\mathbf{B}^{(2)}$ are full-rank. Hence, we can express $\mathbf{B}^{(2)} = (1-4A)(\mathbf{B}^{(1)})^{-\top}$. Also, note that

$$\frac{2}{1-8A} - \frac{1}{1-4A} = \frac{2 - 8A - 1 + 8A}{(1-8A)(1-4A)} = (1-8A)^{-1}(1-4A)^{-1}.$$

We rewrite the expression for the variance using the identity above and the formula for $\mathbf{B}^{(2)}$:

$$\mathrm{Var}_{\nu_{\mathrm{ADE}}} f_{\mathrm{ADE}}^{(1)}(\boldsymbol{\omega}, \mathbf{x}) f_{\mathrm{ADE}}^{(2)}(\boldsymbol{\omega}, \mathbf{y}) = \left(\frac{1-4A}{\sqrt{1-8A}}\right)^d \exp\left((1-8A)^{-1}(1-4A)^{-1}\mathbf{x}^\top(\mathbf{B}^{(1)})^\top \mathbf{B}^{(1)}\mathbf{x}\right.$$

$$\left. + (1-8A)^{-1}(1-4A)\mathbf{y}^\top((\mathbf{B}^{(1)})^\top \mathbf{B}^{(1)})^{-1}\mathbf{y} + 4(1-8A)^{-1}(1-4A)\mathbf{x}^\top \mathbf{y}\right) - K^{(0)}(\mathbf{x}, \mathbf{y})^2.$$

We use the expression above to rewrite (8) for $\langle \nu, f^{(1)}, f^{(2)}\rangle = \langle \nu_{\mathrm{ADE}}, f_{\mathrm{ADE}}^{(1)}, f_{\mathrm{ADE}}^{(2)}\rangle$ as follows:

$$\overline{\mathcal{L}}(\boldsymbol{\theta}_{\mathrm{ADE}}; \mathcal{X}, \mathcal{Y}, \mathcal{T}_{\mathrm{ADE}}) = L^{-2} \sum_{1 \leq i,j \leq L} \log(\mathrm{Var}_{\nu_{\mathrm{ADE}}} f_{\mathrm{ADE}}^{(1)}(\boldsymbol{\omega}, \mathbf{x}^{(i)}) f_{\mathrm{ADE}}^{(2)}(\boldsymbol{\omega}, \mathbf{y}^{(j)}) + K^{(0)}(\mathbf{x}^{(i)}, \mathbf{y}^{(j)}))$$

$$= d\log(1-4A) - \frac{d}{2}\log(1-8A) + (1-8A)^{-1}(1-4A)^{-1}L^{-1}\sum_{i=1}^{L}(\mathbf{x}^{(i)})^\top(\mathbf{B}^{(1)})^\top \mathbf{B}^{(1)}\mathbf{x}^{(i)}$$

$$+ (1-8A)^{-1}(1-4A)L^{-1}\sum_{j=1}^{L}(\mathbf{y}^{(j)})^\top(\mathbf{B}^{(1)})^{-1}(\mathbf{B}^{(1)})^{-\top}\mathbf{y}^{(j)}$$

$$+4(1-8A)^{-1}(1-4A)L^{-2}\sum_{1\leq i,j\leq L}(\mathbf{x}^{(i)})^{\top}\mathbf{y}^{(j)}. \tag{21}$$

Denote $\mathbf{E} = (\mathbf{B}^{(1)})^{\top}\mathbf{B}^{(1)}$. Then (21) becomes:

$$\overline{\mathcal{L}}(\boldsymbol{\theta}_{\mathrm{ADE}}; \mathcal{X}, \mathcal{Y}, \mathcal{T}_{\mathrm{ADE}}) = d\log(1-4A) - \frac{d}{2}\log(1-8A)$$

$$+(1-8A)^{-1}(1-4A)^{-1}L^{-1}\sum_{i=1}^{L}(\mathbf{x}^{(i)})^{\top}\mathbf{E}\mathbf{x}^{(i)}$$

$$+(1-8A)^{-1}(1-4A)L^{-1}\sum_{j=1}^{L}(\mathbf{y}^{(j)})^{\top}\mathbf{E}^{-1}\mathbf{y}^{(j)} + 4(1-8A)^{-1}(1-4A)L^{-2}\sum_{1\leq i,j\leq L}(\mathbf{x}^{(i)})^{\top}\mathbf{y}^{(j)}. \tag{22}$$

We next prove the following lemma:

**Lemma A.2.** *Let* $\mathbf{B}^{(1)*} = \sqrt{1-4A}\boldsymbol{\Sigma}^{1/2}\mathbf{U}^{\top}(\boldsymbol{\Lambda}^{(1)})^{-1/2}(\mathbf{Q}^{(1)})^{\top}$. *When* $A$ *($8A < 1$) is fixed,* $\mathbf{E} = \mathbf{E}^{*} = (\mathbf{B}^{(1)*})^{\top}\mathbf{B}^{(1)*}$ *minimizes the right hand side of (22) with respect to* $\mathbf{E}$.

*Proof.* We have:

$$L^{-1}\sum_{i=1}^{L}(\mathbf{x}^{(i)})^{\top}\mathbf{E}\mathbf{x}^{(i)} = L^{-1}\sum_{i=1}^{L}\mathrm{Trace}((\mathbf{x}^{(i)})^{\top}\mathbf{E}\mathbf{x}^{(i)}) = L^{-1}\sum_{i=1}^{L}\mathrm{Trace}(\mathbf{E}\mathbf{x}^{(i)}(\mathbf{x}^{(i)})^{\top})$$

$$= \mathrm{Trace}\left(\mathbf{E}\left(L^{-1}\sum_{i=1}^{L}\mathbf{x}^{(i)}(\mathbf{x}^{(i)})^{\top}\right)\right) = \mathrm{Trace}(\mathbf{E}\mathbf{M}^{(1)})$$

where we use the cyclic property of trace $\mathrm{Trace}(\cdot)$ and linearity of trace. Analogously, we obtain $L^{-1}\sum_{j=1}^{L}(\mathbf{y}^{(j)})^{\top}\mathbf{E}^{-1}\mathbf{y}^{(j)} = \mathrm{Trace}(\mathbf{E}^{-1}\mathbf{M}^{(2)})$. Assuming that $A$ is fixed, optimization of (22) with respect to $\mathbf{E}$ reduces to the following minimization problem:

$$\min_{\mathbf{E}\in\mathbb{S}_d, \mathbf{E}\succ 0} \mathcal{F}(\mathbf{E}) = \beta_1\mathrm{Trace}(\mathbf{E}\mathbf{M}^{(1)}) + \beta_2\mathrm{Trace}(\mathbf{E}^{-1}\mathbf{M}^{(2)}) \tag{23}$$

where $\beta_1 = (1-8A)^{-1}(1-4A)^{-1}$, $\beta_2 = (1-8A)^{-1}(1-4A)$ and the constraint $\mathbf{E}\in\mathbb{S}_d, \mathbf{E}\succ 0$ follows from the fact that $\mathbf{E} = (\mathbf{B}^{(1)})^{\top}\mathbf{B}^{(1)}$ and $\mathbf{E}$ is invertible. We have $1-8A > 0$ and $1-4A = (1-8A)/2 + 1/2 > 0$. Hence, $\beta_1, \beta_2 > 0$. For any $\mathbf{E}\succ 0$ and any $\boldsymbol{\Delta}\in\mathbb{S}_d$ there is $t\in\mathbb{R}$ small enough such that $\mathbf{E}+t\mathbf{B}$ is invertible and the following Neumann series is convergent:

$$(\mathbf{E}+t\boldsymbol{\Delta})^{-1} = \mathbf{E}^{-1}(\mathbf{I}_d + t\boldsymbol{\Delta}\mathbf{E}^{-1})^{-1} = \sum_{l=0}^{\infty}(-t)^l\mathbf{E}^{-1}(\boldsymbol{\Delta}\mathbf{E}^{-1})^l$$

We further deduce:

$$\mathrm{Trace}((\mathbf{E}+t\boldsymbol{\Delta})^{-1}\mathbf{M}^{(2)}) = \mathrm{Trace}\left(\left(\sum_{l=0}^{\infty}(-t)^l\mathbf{E}^{-1}(\boldsymbol{\Delta}\mathbf{E}^{-1})^l\right)\mathbf{M}^{(2)}\right) =$$

$$= \sum_{l=0}^{\infty}(-t)^l\mathrm{Trace}\left(\mathbf{E}^{-1}(\boldsymbol{\Delta}\mathbf{E}^{-1})^l\mathbf{M}^{(2)}\right)$$

and, therefore,

$$\mathcal{F}(\mathbf{E}+t\boldsymbol{\Delta}) = \beta_1\mathrm{Trace}((\mathbf{E}+t\boldsymbol{\Delta})\mathbf{M}^{(1)}) + \beta_2\sum_{l=0}^{\infty}(-t)^l\mathrm{Trace}\left(\mathbf{E}^{-1}(\boldsymbol{\Delta}\mathbf{E}^{-1})^l\mathbf{M}^{(2)}\right)$$

$$= \beta_1\mathrm{Trace}(\mathbf{E}\mathbf{M}^{(1)}) + t\beta_1\mathrm{Trace}(\boldsymbol{\Delta}\mathbf{M}^{(1)}) + \beta_2\sum_{l=0}^{\infty}(-t)^l\mathrm{Trace}\left(\mathbf{E}^{-1}(\boldsymbol{\Delta}\mathbf{E}^{-1})^l\mathbf{M}^{(2)}\right). \tag{24}$$

Further, we have:

$$\frac{\partial}{\partial t}\mathcal{F}(\mathbf{E}+t\boldsymbol{\Delta}) = \beta_1 \text{Trace}(\boldsymbol{\Delta}\mathbf{M}^{(1)}) + \beta_2 \sum_{l=1}^{\infty}(-1)^l l t^{l-1}\text{Trace}\left(\mathbf{E}^{-1}(\boldsymbol{\Delta}\mathbf{E}^{-1})^l\mathbf{M}^{(2)}\right),$$

$$\frac{\partial^2}{(\partial t)^2}\mathcal{F}(\mathbf{E}+t\boldsymbol{\Delta}) = \beta_2 \sum_{l=2}^{\infty}(-1)^l l(l-1) t^{l-2}\text{Trace}\left(\mathbf{E}^{-1}(\boldsymbol{\Delta}\mathbf{E}^{-1})^l\mathbf{M}^{(2)}\right),$$

$$\left.\frac{\partial^2}{(\partial t)^2}\mathcal{F}(\mathbf{E}+t\boldsymbol{\Delta})\right|_{t=0} = 2\beta_2 \text{Trace}\left(\mathbf{E}^{-1}\boldsymbol{\Delta}\mathbf{E}^{-1}\boldsymbol{\Delta}\mathbf{E}^{-1}\mathbf{M}^{(2)}\right). \tag{25}$$

We replace $\mathbf{M}^{(2)} = \mathbf{Q}^{(2)}(\boldsymbol{\Lambda}^{(2)})^{1/2}(\boldsymbol{\Lambda}^{(2)})^{1/2}(\mathbf{Q}^{(2)})^{\top}$ and apply the cyclic property of trace in (25):

$$\left.\frac{\partial^2}{(\partial t)^2}\mathcal{F}(\mathbf{E}+t\boldsymbol{\Delta})\right|_{t=0} = 2\beta_2 \text{Trace}\left((\boldsymbol{\Lambda}^{(2)})^{1/2}(\mathbf{Q}^{(2)})^{\top}\mathbf{E}^{-1}\boldsymbol{\Delta}\mathbf{E}^{-1}\boldsymbol{\Delta}\mathbf{E}^{-1}\mathbf{Q}^{(2)}(\boldsymbol{\Lambda}^{(2)})^{1/2}\right)$$

$$= 2\beta_2 \text{Trace}\left(\mathbf{T}\mathbf{E}^{-1}\mathbf{T}^{\top}\right)$$

where $\mathbf{T} = (\boldsymbol{\Lambda}^{(2)})^{1/2}(\mathbf{Q}^{(2)})^{\top}\mathbf{E}^{-1}\boldsymbol{\Delta}$. Since $\mathbf{E}$ is positive definite, $\mathbf{E}^{-1}$ is also positive definite and $\mathbf{T}\mathbf{E}^{-1}\mathbf{T}^{\top}$ is at least positive semidefinite. Hence, $\text{Trace}\left(\mathbf{T}\mathbf{E}^{-1}\mathbf{T}^{\top}\right) \geq 0$ and also $\frac{\partial^2}{(\partial t)^2}\mathcal{F}(\mathbf{E}+t\boldsymbol{\Delta})|_{t=0} \geq 0$. We conclude that $\mathcal{F}(\mathbf{E})$ is a convex function on $\{\mathbf{E} \in \mathbb{S}_d \,|\, \mathbf{E} \succ 0\}$. Since $\{\mathbf{E} \in \mathbb{S}_d \,|\, \mathbf{E} \succ 0\}$ is an open set, (every) global minimum $\mathbf{E}$ of (23) satisfies two conditions

$$1)\ \mathbf{E} \succ 0, \quad \text{and} \quad 2)\ \nabla\mathcal{F}(\mathbf{E}) = \mathbf{0}_{d\times d} \tag{26}$$

Set $t = 1$ and assume that $\boldsymbol{\Delta} \in \mathbb{S}_d$ is small enough by norm so that $\mathbf{E}+\boldsymbol{\Delta}$ is invertible and the Neumann series for $(\mathbf{I}_d + \boldsymbol{\Delta}\mathbf{E}^{-1})^{-1}$ is convergent. Then, (24) holds for $t = 1$:

$$\mathcal{F}(\mathbf{E}+\boldsymbol{\Delta}) = \beta_1 \text{Trace}(\mathbf{E}\mathbf{M}^{(1)}) + \beta_1 \text{Trace}(\boldsymbol{\Delta}\mathbf{M}^{(1)}) + \beta_2 \sum_{l=0}^{\infty}(-1)^l \text{Trace}\left(\mathbf{E}^{-1}(\boldsymbol{\Delta}\mathbf{E}^{-1})^l\mathbf{M}^{(2)}\right)$$

$$= \mathcal{F}(\mathbf{E}) + \beta_1 \text{Trace}(\boldsymbol{\Delta}\mathbf{M}^{(1)}) + \beta_2 \sum_{l=1}^{\infty}(-1)^l \text{Trace}\left(\mathbf{E}^{-1}(\boldsymbol{\Delta}\mathbf{E}^{-1})^l\mathbf{M}^{(2)}\right)$$

$$= \mathcal{F}(\mathbf{E}) + \beta_1 \text{Trace}(\boldsymbol{\Delta}\mathbf{M}^{(1)}) - \beta_2 \text{Trace}\left(\mathbf{E}^{-1}\boldsymbol{\Delta}\mathbf{E}^{-1}\mathbf{M}^{(2)}\right)$$

$$+ \beta_2 \sum_{l=2}^{\infty}(-1)^l \text{Trace}\left(\mathbf{E}^{-1}(\boldsymbol{\Delta}\mathbf{E}^{-1})^l\mathbf{M}^{(2)}\right).$$

Clearly, $\beta_2 \sum_{l=2}^{\infty}(-1)^l \text{Trace}\left(\mathbf{E}^{-1}(\boldsymbol{\Delta}\mathbf{E}^{-1})^l\mathbf{M}^{(2)}\right) = o(\|\boldsymbol{\Delta}\|)$ where $\|\cdot\|$ is an $L_2$-norm. Also, using the cyclic property of trace, we get:

$$\text{Trace}\left(\mathbf{E}^{-1}\boldsymbol{\Delta}\mathbf{E}^{-1}\mathbf{M}^{(2)}\right) = \text{Trace}\left(\boldsymbol{\Delta}\mathbf{E}^{-1}\mathbf{M}^{(2)}\mathbf{E}^{-1}\right).$$

Therefore, we have:

$$\mathcal{F}(\mathbf{E}+\boldsymbol{\Delta}) = \text{Trace}\left(\boldsymbol{\Delta}\left(\beta_1\mathbf{M}^{(1)} - \beta_2\mathbf{E}^{-1}\mathbf{M}^{(2)}\mathbf{E}^{-1}\right)\right) + o(\|\boldsymbol{\Delta}\|). \tag{27}$$

Since $\boldsymbol{\Delta}, \mathbf{E}^{-1}, \mathbf{M}^{(1)}, \mathbf{M}^{(2)} \in \mathbb{S}_d$, from (27) it follows that

$$\nabla\mathcal{F}(\mathbf{E}) = \beta_1\mathbf{M}^{(1)} - \beta_2\mathbf{E}^{-1}\mathbf{M}^{(2)}\mathbf{E}^{-1}. \tag{28}$$

Let $\mathbf{E}^* = (\mathbf{B}^{(1)*})^{\top}\mathbf{B}^{(1)*} \succeq 0$. Note that

$$\sqrt[4]{\frac{\beta_2}{\beta_1}} = \sqrt[4]{\frac{(1-8A)^{-1}(1-4A)}{(1-8A)^{-1}(1-4A)^{-1}}} = \sqrt{1-4A}.$$

Since $\sqrt{\beta_2/\beta_1} \neq 0$, $\boldsymbol{\Sigma}, \mathbf{U}, \boldsymbol{\Lambda}^{-1/2}, \mathbf{Q}^{(1)}$ are full-rank, $\mathbf{E}^*$ is also full-rank, therefore $\mathbf{E}^* \succ 0$ and it satisfies condition 1 from (26). Observe that

$$\mathbf{E}^*\mathbf{Q}^{(1)}(\boldsymbol{\Lambda}^{(1)})^{1/2} = \sqrt{\beta_2/\beta_1}\mathbf{Q}^{(1)}(\boldsymbol{\Lambda}^{(1)})^{-1/2}\mathbf{U}\boldsymbol{\Sigma}\mathbf{U}^{\top}(\boldsymbol{\Lambda}^{(1)})^{-1/2}(\mathbf{Q}^{(1)})^{\top}\mathbf{Q}^{(1)}(\boldsymbol{\Lambda}^{(1)})^{1/2}$$

$$= \sqrt{\beta_2/\beta_1}\mathbf{Q}^{(1)}(\mathbf{\Lambda}^{(1)})^{-1/2}\mathbf{U}\mathbf{\Sigma}\mathbf{U}^\top$$
$$= \sqrt{\beta_2/\beta_1}\mathbf{Q}^{(1)}(\mathbf{\Lambda}^{(1)})^{-1/2}(\mathbf{U}\mathbf{\Sigma}\mathbf{V}^\top)\mathbf{V}\mathbf{U}^\top$$
$$= \sqrt{\beta_2/\beta_1}\mathbf{Q}^{(1)}(\mathbf{\Lambda}^{(1)})^{-1/2}((\mathbf{\Lambda}^{(1)})^{\frac{1}{2}}(\mathbf{Q}^{(1)})^\top\mathbf{Q}^{(2)}(\mathbf{\Lambda}^{(2)})^{\frac{1}{2}})\mathbf{V}\mathbf{U}^\top$$
$$= \sqrt{\beta_2/\beta_1}\mathbf{Q}^{(2)}(\mathbf{\Lambda}^{(2)})^{1/2}\mathbf{V}\mathbf{U}^\top$$

where we use definitions of $\mathbf{E}^*$, $\mathbf{U}$, $\mathbf{\Sigma}$, $\mathbf{V}$ and orthogonality of $\mathbf{Q}^{(1)}$, $\mathbf{Q}^{(2)}$, $\mathbf{U}$, $\mathbf{V}$. Hence, we deduce that

$$\beta_1\mathbf{E}^*\mathbf{M}^{(1)}\mathbf{E}^* = \beta_1\mathbf{E}^*\mathbf{Q}^{(1)}(\mathbf{\Lambda}^{(1)})^{1/2}\left((\mathbf{\Lambda}^{(1)})^{1/2}(\mathbf{Q}^{(1)})^\top\mathbf{E}^*\right) \tag{29}$$
$$= \beta_1\frac{\beta_2}{\beta_1}\mathbf{Q}^{(2)}(\mathbf{\Lambda}^{(2)})^{1/2}\left((\mathbf{\Lambda}^{(2)})^{1/2}(\mathbf{Q}^{(2)})^\top\right)$$
$$= \beta_2\mathbf{M}^{(2)} \tag{30}$$

by the definition of $\mathbf{Q}^{(1)}$, $\mathbf{\Lambda}^{(1)}$, $\mathbf{Q}^{(2)}$, $\mathbf{\Lambda}^{(2)}$ and due to orthogonality of $\mathbf{V}$, $\mathbf{U}$. By left- and right-multiplication of (30) by $(\mathbf{E}^*)^{-1}$ we deduce that

$$\beta_1\mathbf{M}^{(1)} = \beta_2(\mathbf{E}^*)^{-1}\mathbf{M}^{(2)}(\mathbf{E}^*)^{-1}$$

or, in other words, $\nabla\mathcal{F}(\mathbf{E}^*) = \mathbf{0}_{d\times d}$ and the condition 2 from (26) is also satisfied. We conclude that the global minimum of (23) is achieved at $\mathbf{E}^*$. $\qquad\square$

According to Lemma A.2, $\mathbf{B}^{(1)} = \mathbf{B}^{(1)*}$ is a global minimum of (21)'s right hand side when $A$ is fixed. Indeed, if there is $\mathbf{B}^{(1)}$ which leads to a smaller value of (21), $\mathbf{E} = (\mathbf{B}^{(1)})^\top\mathbf{B}^{(1)}$ would lead to a smaller value of (22)'s right hand side. Also, this $\mathbf{E}$ is positive definite by definition (note that $\mathbf{B}^{(1)}$ is nonsingular), leading to contradiction with Lemma A.2.

Substituting $\mathbf{E}^*$ instead of $\mathbf{E}$ in (22) corresponds to the minimum value of $\overline{\mathcal{L}}(\boldsymbol{\theta}_{\text{AGE}};\alpha,\mathcal{X},\mathcal{Y},\mathcal{T}_{\text{AGE}})$ for a fixed $A$. Our next step is to minimize this expression with respect to $A$. Denote $\mathbf{F} = \mathbf{Q}^{(1)}(\mathbf{\Lambda}^{(1)})^{-1/2}\mathbf{U}\mathbf{\Sigma}\mathbf{U}^\top(\mathbf{\Lambda}^{(1)})^{-1/2}(\mathbf{Q}^{(1)})^\top$. Then $\mathbf{E}^* = (1-4A)\mathbf{F}$ where $\mathbf{F}$ doesn't depend on $A$. We substitute $\mathbf{E}^*$ into (22) and get:

$$d\log(1-4A) - \frac{d}{2}\log(1-8A) + (1-8A)^{-1}(1-4A)^{-1}\text{Trace}((1-4A)\mathbf{F}\mathbf{M}^{(1)})$$
$$+(1-8A)^{-1}(1-4A)\text{Trace}((1-4A)^{-1}\mathbf{F}^{-1}\mathbf{M}^{(2)})$$
$$+4(1-8A)^{-1}(1-4A)L^{-2}\sum_{1\le i,j\le L}(\mathbf{x}^{(i)})^\top\mathbf{y}^{(j)}$$

$$= d\log(1-4A) - \frac{d}{2}\log(1-8A) + (1-8A)^{-1}\text{Trace}(\mathbf{F}\mathbf{M}^{(1)}) + (1-8A)^{-1}\text{Trace}(\mathbf{F}^{-1}\mathbf{M}^{(2)})$$
$$+2\left(1+(1-8A)^{-1}\right)d\mu^{(3)} \tag{31}$$

where we also replace

$$L^{-2}\sum_{1\le i,j\le L}(\mathbf{x}^{(i)})^\top\mathbf{y}^{(j)} = L^{-2}\left(\sum_{i=1}^{L}\mathbf{x}^{(i)}\right)^\top\left(\sum_{j=1}^{L}\mathbf{y}^{(j)}\right) = d\mu^{(3)}$$

and

$$(1-8A)^{-1}(1-4A) = \frac{(1-8A)+1}{2(1-8A)} = \frac{1}{2}\left(1+(1-8A)^{-1}\right)$$

Based on (30) and since $\mathbf{F} = \sqrt{\beta_1/\beta_2}\mathbf{E}^*$, we conclude that $\mathbf{F}\mathbf{M}^{(1)}\mathbf{F} = \mathbf{M}^{(2)}$, or $\mathbf{M}^{(1)}\mathbf{F} = \mathbf{F}^{-1}\mathbf{M}^{(2)}$. Using the cyclic property of trace, we get:

$$\text{Trace}(\mathbf{F}\mathbf{M}^{(1)}) = \text{Trace}(\mathbf{M}^{(1)}\mathbf{F}) = \text{Trace}(\mathbf{F}^{-1}\mathbf{M}^{(2)}).$$

By the definition of $\mathbf{F}$, $\mathbf{\Lambda}^{(1)}$, $\mathbf{Q}^{(1)}$ and using the cyclic property and orthogonality of $\mathbf{Q}^{(1)}$, $\mathbf{U}$, we have:

$$\text{Trace}(\mathbf{F}\mathbf{M}^{(1)}) = \text{Trace}\left(\mathbf{Q}^{(1)}(\mathbf{\Lambda}^{(1)})^{-1/2}\mathbf{U}\mathbf{\Sigma}\mathbf{U}^\top(\mathbf{\Lambda}^{(1)})^{-1/2}\mathbf{Q}^{(1)})^\top\left(\mathbf{Q}^{(1)}\mathbf{\Lambda}^{(1)}(\mathbf{Q}^{(1)})^\top\right)\right)$$

$$= \text{Trace}\left(\mathbf{Q}^{(1)}(\mathbf{\Lambda}^{(1)})^{-1/2}\mathbf{U}\mathbf{\Sigma}\mathbf{U}^\top(\mathbf{\Lambda}^{(1)})^{1/2}(\mathbf{Q}^{(1)})^\top\right)$$

$$= \text{Trace}\left(\mathbf{\Sigma}\mathbf{U}^\top(\mathbf{\Lambda}^{(1)})^{1/2}(\mathbf{Q}^{(1)})^\top\mathbf{Q}^{(1)}(\mathbf{\Lambda}^{(1)})^{-1/2}\mathbf{U}\right)$$

$$= \text{Trace}(\mathbf{\Sigma}) = \sum_{l=1}^{d}\mathbf{\Sigma}_{l,l}.$$

Hence, (31) finally becomes:

$$d\log(1-4A) - \frac{d}{2}\log(1-8A) + 2(1-8A)^{-1}\sum_{l=1}^{d}\mathbf{\Sigma}_{l,l} + 2\left(1+(1-8A)^{-1}\right)d\mu^{(3)}$$

$$= d\left(\log(1-4A) - \frac{1}{2}\log(1-8A) + 2(1-8A)^{-1}\left(d^{-1}\sum_{l=1}^{d}\mathbf{\Sigma}_{l,l} + \mu^{(3)}\right) + 2\mu^{(3)}\right). \quad (32)$$

Next, we use Lemma A.1 ($\phi = d^{-1}\sum_{l=1}^{d}\mathbf{\Sigma}_{l,l} + \mu^{(3)} \geq 0$) for deriving expression for $A$ which minimizes (32). This expression coincides with the one in Theorem's statement. The expressions for $\mathbf{B}^{(2)}, \mathbf{C}^{(1)}, \mathbf{C}^{(2)}$ follow directly from (20), optimal $\mathbf{B}^{(1)} = \mathbf{B}^{(1)*}$ and $A$. (10) follows from (32). The proof is concluded. $\quad\square$

### A.4  Proof of Theorem 4.3

*Proof.* With $\mathbf{B}^{(1)} = \mathbf{B}^{(2)} = \mathbf{B}$ and $\mathbf{C}^{(1)} = \mathbf{C}^{(2)} = \mathbf{C}$, the conditions from Theorem 4.1 read as

$$8A \prec \mathbf{I}_d, \quad \mathbf{B}^\top(\mathbf{I}_d-4A)^{-1}\mathbf{B} = \mathbf{I}_d, \quad \mathbf{C} = -\frac{1}{2}\mathbf{B}^\top(\mathbf{I}_d-4A)^{-1}\mathbf{B} = -\frac{1}{2}\mathbf{I}_d, \quad D = \det(\mathbf{I}_d-4A)^{1/4}. \quad (33)$$

Denote $\mathbf{Q} = (\mathbf{I}_d - 4A)^{-1/2}\mathbf{B} \in \mathbb{R}^{d\times d}$. Then, according to (33), $\mathbf{Q}^\top\mathbf{Q} = \mathbf{I}_d$, that is $\mathbf{Q} \in \mathbb{O}_d$. We rewrite (9) using (33) and then substitute $\mathbf{B} = (\mathbf{I}_d - 4A)^{1/2}\mathbf{Q}$:

$$\text{Var}_{\nu_{\text{SDE}}}f_{\text{SDE}}^{(1)}(\boldsymbol{\omega}, \mathbf{x})f_{\text{SDE}}^{(2)}(\boldsymbol{\omega}, \mathbf{y}) = \det(\mathbf{I}_d - 4A)\det(\mathbf{I}_d - 8A)^{-1/2}\exp\left(-\|\mathbf{x}\|^2\right.$$

$$\left.+2\mathbf{x}^\top\mathbf{B}^\top(\mathbf{I}_d - 8A)^{-1}\mathbf{B}\mathbf{x} - \|\mathbf{y}\|^2 + 2\mathbf{y}^\top\mathbf{B}^\top(\mathbf{I}_d - 8A)^{-1}\mathbf{B}\mathbf{y} + 4\mathbf{x}^\top\mathbf{B}^\top(\mathbf{I}_d - 8A)^{-1}\mathbf{B}\mathbf{y}\right)$$

$$-K^{(0)}(\mathbf{x}, \mathbf{y})^2 = \det(\mathbf{I}_d - 4A)^{1/4}\det(\mathbf{I}_d - 8A)^{-1/2}\exp\left(-\|\mathbf{x}\|^2 - 2\mathbf{x}^\top\mathbf{Q}^\top\mathbf{E}\mathbf{Q}\mathbf{x} - \|\mathbf{y}\|^2\right.$$

$$\left.-2\mathbf{y}^\top\mathbf{Q}^\top\mathbf{E}\mathbf{Q}\mathbf{y} - 4\mathbf{x}^\top\mathbf{Q}^\top\mathbf{E}\mathbf{Q}\mathbf{y}\right) - K^{(0)}(\mathbf{x}, \mathbf{y})^2 \quad (34)$$

where we denote:

$$\mathbf{E} = -(\mathbf{I}_d - 4A)^{1/2}(\mathbf{I}_d - 8A)^{-1}(\mathbf{I}_d - 4A)^{1/2} = -(\mathbf{I}_d - 4A)(\mathbf{I}_d - 8A)^{-1}$$

$$= -\frac{1}{2}\left((\mathbf{I}_d - 8A) + \mathbf{I}_d\right)(\mathbf{I}_d - 8A)^{-1} = -\frac{1}{2}\mathbf{I}_d - \frac{1}{2}(\mathbf{I}_d - 8A)^{-1} \quad (35)$$

which is in $\mathbb{D}_d$ since $A \in \mathbb{D}_d$. Next, we observe:

$$2\mathbf{x}^\top\mathbf{Q}^\top\mathbf{E}\mathbf{Q}\mathbf{x} + 2\mathbf{y}^\top\mathbf{Q}^\top\mathbf{E}\mathbf{Q}\mathbf{y} + 4\mathbf{x}^\top\mathbf{Q}^\top\mathbf{E}\mathbf{Q}\mathbf{y} = 2(\mathbf{x} + \mathbf{y})^\top\mathbf{Q}^\top\mathbf{E}\mathbf{Q}(\mathbf{x} + \mathbf{y})$$

We plug this into (34) and use the resulting expression to rewrite (8) for $\langle \nu, f^{(1)}, f^{(2)}\rangle = \langle \nu_{\text{SDE}}, f_{\text{SDE}}^{(1)}, f_{\text{SDE}}^{(2)}\rangle$ as follows:

$$\overline{\mathcal{L}}(\boldsymbol{\theta}_{\text{SDE}}; \mathcal{X}, \mathcal{Y}, \mathcal{T}_{\text{SDE}}) = L^{-2}\sum_{1\leq i,j\leq L}\log(\text{Var}_{\nu_{\text{SDE}}}f_{\text{SDE}}^{(1)}(\boldsymbol{\omega}, \mathbf{x}^{(i)})f_{\text{SDE}}^{(2)}(\boldsymbol{\omega}, \mathbf{y}^{(j)}) + K^{(0)}(\mathbf{x}^{(i)}, \mathbf{y}^{(j)}))$$

$$= \log\det(\mathbf{I}_d - 4A) - \frac{1}{2}\log\det(\mathbf{I}_d - 8A) - L^{-1}\sum_{i=1}^{L}\|\mathbf{x}^{(i)}\|^2 - L^{-1}\sum_{j=1}^{L}\|\mathbf{y}^{(j)}\|^2$$

$$-2L^{-2}\sum_{1\leq i,j\leq L}(\mathbf{x}^{(i)} + \mathbf{y}^{(j)})^\top\mathbf{Q}^\top\mathbf{E}\mathbf{Q}(\mathbf{x}^{(i)} + \mathbf{y}^{(j)}). \quad (36)$$

Using linearity and cyclic property of trace, we deduce that

$$L^{-2} \sum_{1 \le i,j \le L} (\mathbf{x}^{(i)} + \mathbf{y}^{(j)})^\top \mathbf{Q}^\top \mathbf{E} \mathbf{Q} (\mathbf{x}^{(i)} + \mathbf{y}^{(j)}) =$$

$$= L^{-2} \sum_{1 \le i,j \le L} \mathrm{Trace}\left( (\mathbf{x}^{(i)} + \mathbf{y}^{(j)})^\top \mathbf{Q}^\top \mathbf{E} \mathbf{Q} (\mathbf{x}^{(i)} + \mathbf{y}^{(j)}) \right)$$

$$= L^{-2} \sum_{1 \le i,j \le L} \mathrm{Trace}\left( \mathbf{Q}^\top \mathbf{E} \mathbf{Q} (\mathbf{x}^{(i)} + \mathbf{y}^{(j)})(\mathbf{x}^{(i)} + \mathbf{y}^{(j)})^\top \right)$$

$$= \mathrm{Trace}\left( \mathbf{Q}^\top \mathbf{E} \mathbf{Q} \left( L^{-2} \sum_{1 \le i,j \le L} (\mathbf{x}^{(i)} + \mathbf{y}^{(j)})(\mathbf{x}^{(i)} + \mathbf{y}^{(j)})^\top \right) \right)$$

Observe that

$$L^{-2} \sum_{1 \le i,j \le L} (\mathbf{x}^{(i)} + \mathbf{y}^{(j)})(\mathbf{x}^{(i)} + \mathbf{y}^{(j)})^\top =$$

$$= L^{-2} \sum_{1 \le i,j \le L} \left( \mathbf{x}^{(i)}(\mathbf{x}^{(i)})^\top + \mathbf{x}^{(i)}(\mathbf{y}^{(j)})^\top + \mathbf{y}^{(j)}(\mathbf{x}^{(i)})^\top + \mathbf{y}^{(j)}(\mathbf{x}^{(j)})^\top \right)$$

$$= L^{-1} \sum_{i=1}^{L} \mathbf{x}^{(i)}(\mathbf{x}^{(i)})^\top + \left( L^{-1} \sum_{i=1}^{L} \mathbf{x}^{(i)} \right)\left( L^{-1} \sum_{j=1}^{L} \mathbf{y}^{(j)} \right)^\top + \left( L^{-1} \sum_{j=1}^{L} \mathbf{y}^{(j)} \right)\left( L^{-1} \sum_{i=1}^{L} \mathbf{x}^{(i)} \right)^\top$$

$$+ L^{-1} \sum_{j=1}^{L} \mathbf{y}^{(j)}(\mathbf{y}^{(j)})^\top = \mathbf{M}^{(1)} + \boldsymbol{\mu}^{(4)}(\boldsymbol{\mu}^{(5)})^\top + \boldsymbol{\mu}^{(5)}(\boldsymbol{\mu}^{(4)})^\top + \mathbf{M}^{(2)}.$$

Denote $\mathbf{N} = \mathbf{M}^{(1)} + \boldsymbol{\mu}^{(4)}(\boldsymbol{\mu}^{(5)})^\top + \boldsymbol{\mu}^{(5)}(\boldsymbol{\mu}^{(4)})^\top + \mathbf{M}^{(2)}$. We conclude that

$$\overline{\mathcal{L}}(\boldsymbol{\theta}_{\mathrm{SDE}}; \mathcal{X}, \mathcal{Y}, \mathcal{T}_{\mathrm{SDE}}) = \log \det(\mathbf{I}_d - 4\mathbf{A}) - \frac{1}{2} \log \det(\mathbf{I}_d - 8\mathbf{A}) - L^{-1} \sum_{i=1}^{L} \|\mathbf{x}^{(i)}\|^2$$

$$- L^{-1} \sum_{j=1}^{L} \|\mathbf{y}^{(j)}\|^2 - 2\mathrm{Trace}\left( \mathbf{Q}^\top \mathbf{E} \mathbf{Q} \mathbf{N} \right). \tag{37}$$

With $\mathbf{A}$ fixed, we minimize the right hand side of (37) with respect to $\mathbf{Q}$ which is equivalent to minimizing $\overline{\mathcal{L}}(\boldsymbol{\theta}_{\mathrm{SDE}}; \mathcal{X}, \mathcal{Y}, \mathcal{T}_{\mathrm{SDE}})$ with respect to $\mathbf{B}$ with fixed $\mathbf{A}$, since there is a one-to-one correspondence between $\mathbf{B}$ and $\mathbf{Q}$. This is equivalent to maximizing, again using the cyclic property of trace,

$$\mathrm{Trace}\left( \mathbf{Q}^\top \mathbf{E} \mathbf{Q} \mathbf{N} \right) = \mathrm{Trace}\left( \mathbf{E} \mathbf{Q} \mathbf{N} \mathbf{Q}^\top \right) \tag{38}$$

with respect to $\mathbf{Q}$. We prove the following lemma first:

**Lemma A.3.** *Suppose that diagonal entries of $\mathbf{E}$ are all distinct, and the same holds for $\mathbf{\Lambda}^{(3)}$. Let $\mathbf{\Pi} \in \{0,1\}^{d \times d}$ be a permutation matrix sorting diagonal entries of $\mathbf{E}$ (i.e. by applying $\mathbf{\Pi} \mathbf{E} \mathbf{\Pi}^\top$) in a descending order corresponding to a permutation $\boldsymbol{\pi} \in \mathbb{N}^d$. Set $\mathbf{Q}^* = \mathbf{\Pi}^\top (\mathbf{Q}^{(3)})^\top \in \mathbb{O}_d$. Then we have:*

$$\mathrm{Trace}\left( \mathbf{E} \mathbf{Q}^* \mathbf{N} (\mathbf{Q}^*)^\top \right) = \sum_{l=1}^{d} \mathbf{E}_{\boldsymbol{\pi}_l, \boldsymbol{\pi}_l} \mathbf{\Lambda}^{(3)}_{l,l} \tag{39}$$

$$= \sup_{\mathbf{Q} \in \mathbb{O}_d} \mathrm{Trace}\left( \mathbf{E} \mathbf{Q} \mathbf{N} \mathbf{Q}^\top \right) \tag{40}$$

*Proof.* First of all, we have:

$$\mathrm{Trace}\left( \mathbf{E} \mathbf{Q}^* \mathbf{N} (\mathbf{Q}^*)^\top \right) = \mathrm{Trace}\left( \mathbf{E} \mathbf{\Pi}^\top (\mathbf{Q}^{(3)})^\top \mathbf{N} \mathbf{Q}^{(3)} \mathbf{\Pi} \right) = \mathrm{Trace}\left( \mathbf{E} \mathbf{\Pi}^\top \mathbf{\Lambda}^{(3)} \mathbf{\Pi} \right) \tag{41}$$

$$= \text{Trace}\left(\mathbf{\Pi}\mathbf{E}\mathbf{\Pi}^\top\mathbf{\Lambda}^{(3)}\right) = \sum_{l=1}^d \mathbf{E}_{\boldsymbol{\pi}_l,\boldsymbol{\pi}_l}\mathbf{\Lambda}_{l,l}^{(3)}, \tag{42}$$

i.e. (39) is satisfied.

Optimization for finding $\sup_{\mathbf{Q}\in\mathbb{O}_d}\text{Trace}\left(\mathbf{E}\mathbf{Q}\mathbf{N}\mathbf{Q}^\top\right)$ is a well-studied problem [9]. By the definition, $\mathbf{\Lambda}^{(3)}$ has eigenvalues of $\mathbf{N}$ on the main diagonal and $\mathbf{E} \in \mathbb{D}_d$ hence it contains its eigenvalues on its main diagonal. Then, as proven in [9], $\mathbf{Q}^*$ is indeed a global maximum of this problem in the case of distinct eigenvalues for $\mathbf{E}$ and $\mathbf{N}$. That is, (40) is proven. $\qquad\square$

Next, we prove a generalization of Lemma A.3 when diagonal entries of $\mathbf{E}$ and $\mathbf{\Lambda}^{(3)}$ are not necessarily distinct:

**Lemma A.4.** *Let $\mathbf{\Pi} \in \{0,1\}^{d\times d}$ be a permutation matrix sorting diagonal entries of $\mathbf{E}$ (i.e. by applying $\mathbf{\Pi}\mathbf{E}\mathbf{\Pi}^\top$) in **any non-ascending** order corresponding to a permutation $\boldsymbol{\pi} \in \mathbb{N}^d$. Set $\mathbf{Q}^* = \mathbf{\Pi}^\top(\mathbf{Q}^{(3)})^\top \in \mathbb{O}_d$. Then we have:*

$$\text{Trace}\left(\mathbf{E}\mathbf{Q}^*\mathbf{N}(\mathbf{Q}^*)^\top\right) = \sum_{l=1}^d \mathbf{E}_{\boldsymbol{\pi}_l,\boldsymbol{\pi}_l}\mathbf{\Lambda}_{l,l}^{(3)} \tag{43}$$

$$= \sup_{\mathbf{Q}\in\mathbb{O}_d}\text{Trace}\left(\mathbf{E}\mathbf{Q}\mathbf{N}\mathbf{Q}^\top\right) \tag{44}$$

*Proof.* In the same way as (41-42), we show that $\text{Trace}\left(\mathbf{E}\mathbf{Q}^*\mathbf{N}(\mathbf{Q}^*)^\top\right) = \sum_{l=1}^d \mathbf{E}_{\boldsymbol{\pi}_l,\boldsymbol{\pi}_l}\mathbf{\Lambda}_{l,l}^{(3)}$, i.e. (43) is satisfied. Next we prove that for any $\mathbf{Q} \in \mathbb{O}_d$,

$$\text{Trace}\left(\mathbf{E}\mathbf{Q}\mathbf{N}\mathbf{Q}^\top\right) \leq \sum_{l=1}^d \mathbf{E}_{\boldsymbol{\pi}_l,\boldsymbol{\pi}_l}\mathbf{\Lambda}_{l,l}^{(3)}. \tag{45}$$

which would imply (44).

Our proof is by contradiction. First of all, we can assume that $\mathbf{E}, \mathbf{\Lambda}^{(3)}$ are nonzero matrices since otherwise we have (44) trivially. Since $\text{Trace}\left(\mathbf{E}\mathbf{Q}\mathbf{N}\mathbf{Q}^\top\right)$ is a continuous function of $\mathbf{Q}$ and $\mathbb{O}_d$ is compact, $\sup_{\mathbf{Q}\in\mathbb{O}_d}\text{Trace}\left(\mathbf{E}\mathbf{Q}\mathbf{N}\mathbf{Q}^\top\right)$ is finite. Suppose that there is $\delta > 0$ such that

$$\delta = \sup_{\mathbf{Q}\in\mathbb{O}_d}\text{Trace}\left(\mathbf{E}\mathbf{Q}\mathbf{N}\mathbf{Q}^\top\right) - \sum_{l=1}^d \mathbf{E}_{\boldsymbol{\pi}_l,\boldsymbol{\pi}_l}\mathbf{\Lambda}_{l,l}^{(3)}. \tag{46}$$

Let $\widetilde{\mathbf{E}}, \widetilde{\mathbf{\Lambda}}^{(3)} \in \mathbb{D}_d$ be matrices with all distinct values on the diagonal such that

$$\|\widetilde{\mathbf{E}} - \mathbf{E}\|_\text{F} \leq \min\left(\|\mathbf{E}\|_\text{F}, \frac{\delta}{12\|\mathbf{\Lambda}^{(3)}\|_\text{F}}\right), \quad \|\widetilde{\mathbf{\Lambda}}^{(3)} - \mathbf{\Lambda}^{(3)}\|_\text{F} \leq \frac{\delta}{12\|\mathbf{E}\|_\text{F}} \tag{47}$$

where $\|\cdot\|_\text{F}$ denotes Frobenius norm and $\|\mathbf{E}\|_\text{F}, \|\mathbf{\Lambda}^{(3)}\|_\text{F} \neq 0$ since these are nonzero matrices. Further, we assume that diagonal entries of $\widetilde{\mathbf{\Lambda}}^{(3)}$ are sorted in a descending order and, in addition to $\mathbf{\Lambda}^{(3)}$, $\boldsymbol{\pi}$ also sorts entries of $\widetilde{\mathbf{E}}$ in a non-ascending (descending) order. Clearly, such $\widetilde{\mathbf{E}}, \widetilde{\mathbf{\Lambda}}^{(3)}$ can be obtained by small perturbations of $\mathbf{E}, \mathbf{\Lambda}^{(3)}$. Also, denote $\widetilde{\mathbf{N}} = \mathbf{Q}^{(3)}\widetilde{\mathbf{\Lambda}}^{(3)}(\mathbf{Q}^{(3)})^\top$. Since $\mathbb{O}_d$ is a compact closed set and $\text{Trace}\left(\mathbf{E}\mathbf{Q}\mathbf{N}\mathbf{Q}^\top\right)$ is a continuous function of $\mathbf{Q}$, there exists $\mathbf{Q}^{**} \in \mathbb{O}_d$ such that

$$\text{Trace}\left(\mathbf{E}\mathbf{Q}^{**}\mathbf{N}(\mathbf{Q}^{**})^\top\right) = \sup_{\mathbf{Q}\in\mathbb{O}_d}\text{Trace}\left(\mathbf{E}\mathbf{Q}\mathbf{N}\mathbf{Q}^\top\right). \tag{48}$$

By the definition of $\widetilde{\mathbf{E}}, \widetilde{\mathbf{\Lambda}}^{(3)}, \widetilde{\mathbf{N}}$, we have:

$$\text{Trace}\left(\mathbf{E}\mathbf{Q}^{**}\mathbf{N}(\mathbf{Q}^{**})^\top\right) - \text{Trace}\left(\widetilde{\mathbf{E}}\mathbf{Q}^{**}\widetilde{\mathbf{N}}(\mathbf{Q}^{**})^\top\right)$$

$$= \left(\text{Trace}\left(\mathbf{E}\mathbf{Q}^{**}\mathbf{N}(\mathbf{Q}^{**})^\top\right) - \text{Trace}\left(\mathbf{E}\mathbf{Q}^{**}\widetilde{\mathbf{N}}(\mathbf{Q}^{**})^\top\right)\right) + \left(\text{Trace}\left(\mathbf{E}\mathbf{Q}^{**}\widetilde{\mathbf{N}}(\mathbf{Q}^{**})^\top\right)\right.$$

$$-\operatorname{Trace}\left(\widetilde{\mathbf{E}}\mathbf{Q}^{**}\widetilde{\mathbf{N}}(\mathbf{Q}^{**})^{\top}\right)$$

$$= \operatorname{Trace}\left(\mathbf{E}\mathbf{Q}^{**}\left(\mathbf{N}-\widetilde{\mathbf{N}}\right)(\mathbf{Q}^{**})^{\top}\right) + \operatorname{Trace}\left(\left(\mathbf{E}-\widetilde{\mathbf{E}}\right)\mathbf{Q}^{**}\widetilde{\mathbf{N}}(\mathbf{Q}^{**})^{\top}\right).$$

Next, we apply Cauchy-Schwarz inequality to both terms:

$$\operatorname{Trace}\left(\mathbf{E}\mathbf{Q}^{**}\left(\mathbf{N}-\widetilde{\mathbf{N}}\right)(\mathbf{Q}^{**})^{\top}\right) \leq \|(\mathbf{Q}^{**})^{\top}\mathbf{E}\|_{\mathrm{F}}\|(\mathbf{N}-\widetilde{\mathbf{N}})(\mathbf{Q}^{**})^{\top}\|_{\mathrm{F}} = \|\mathbf{E}\|_{\mathrm{F}}\|\mathbf{N}-\widetilde{\mathbf{N}}\|_{\mathrm{F}},$$

$$\operatorname{Trace}\left(\left(\mathbf{E}-\widetilde{\mathbf{E}}\right)\mathbf{Q}^{**}\widetilde{\mathbf{N}}(\mathbf{Q}^{**})^{\top}\right) \leq \|(\mathbf{Q}^{**})^{\top}(\mathbf{E}-\widetilde{\mathbf{E}})\|_{\mathrm{F}}\|\widetilde{\mathbf{N}}(\mathbf{Q}^{**})^{\top}\|_{\mathrm{F}} = \|\mathbf{E}-\widetilde{\mathbf{E}}\|_{\mathrm{F}}\|\widetilde{\mathbf{N}}\|_{\mathrm{F}}$$

where we use invariance of the Frobenius norm under multiplications by orthogonal matrices. Using this invariance again, we deduce that

$$\|\mathbf{N}-\widetilde{\mathbf{N}}\|_{\mathrm{F}} = \|\mathbf{Q}^{(3)}(\mathbf{\Lambda}^{(3)}-\widetilde{\mathbf{\Lambda}}^{(3)})(\mathbf{Q}^{(3)})^{\top}\|_{\mathrm{F}} = \|\mathbf{\Lambda}^{(3)}-\widetilde{\mathbf{\Lambda}}^{(3)}\|_{\mathrm{F}},$$

$$\|\widetilde{\mathbf{N}}\|_{\mathrm{F}} = \|\mathbf{Q}^{(3)}\widetilde{\mathbf{\Lambda}}^{(3)}(\mathbf{Q}^{(3)})^{\top}\|_{\mathrm{F}} = \|\widetilde{\mathbf{\Lambda}}^{(3)}\|_{\mathrm{F}}.$$

We conclude that

$$\operatorname{Trace}\left(\mathbf{E}\mathbf{Q}^{**}\mathbf{N}(\mathbf{Q}^{**})^{\top}\right) \leq \operatorname{Trace}\left(\widetilde{\mathbf{E}}\mathbf{Q}^{**}\widetilde{\mathbf{N}}(\mathbf{Q}^{**})^{\top}\right) + \|\mathbf{E}\|_{\mathrm{F}}\|\mathbf{\Lambda}^{(3)}-\widetilde{\mathbf{\Lambda}}^{(3)}\|_{\mathrm{F}} + \|\mathbf{E}-\widetilde{\mathbf{E}}\|_{\mathrm{F}}\|\widetilde{\mathbf{\Lambda}}^{(3)}\|_{\mathrm{F}}.$$
(49)

Next, we apply Lemma A.3 to $\mathbf{E}=\widehat{\mathbf{E}}$, $\mathbf{\Lambda}^{(3)}=\widehat{\mathbf{\Lambda}}^{(3)}$ and deduce that

$$\operatorname{Trace}\left(\widetilde{\mathbf{E}}\mathbf{Q}^{**}\widetilde{\mathbf{N}}(\mathbf{Q}^{**})^{\top}\right) \leq \sum_{l=1}^{d}\widetilde{\mathbf{E}}_{\boldsymbol{\pi}_l,\boldsymbol{\pi}_l}\widetilde{\mathbf{\Lambda}}_{l,l}^{(3)} = \sum_{l=1}^{d}\left(\mathbf{E}_{\boldsymbol{\pi}_l,\boldsymbol{\pi}_l}\mathbf{\Lambda}_{l,l}^{(3)} + \left(\widetilde{\mathbf{E}}_{\boldsymbol{\pi}_l,\boldsymbol{\pi}_l} - \mathbf{E}_{\boldsymbol{\pi}_l,\boldsymbol{\pi}_l}\right)\mathbf{\Lambda}_{l,l}^{(3)}\right.$$

$$\left. +\widetilde{\mathbf{E}}_{\boldsymbol{\pi}_l,\boldsymbol{\pi}_l}\left(\widetilde{\mathbf{\Lambda}}_{l,l}^{(3)} - \mathbf{\Lambda}_{l,l}^{(3)}\right)\right)$$

$$= \sum_{l=1}^{d}\mathbf{E}_{\boldsymbol{\pi}_l,\boldsymbol{\pi}_l}\mathbf{\Lambda}_{l,l}^{(3)} + \sum_{l=1}^{d}\left(\widetilde{\mathbf{E}}_{\boldsymbol{\pi}_l,\boldsymbol{\pi}_l} - \mathbf{E}_{\boldsymbol{\pi}_l,\boldsymbol{\pi}_l}\right)\mathbf{\Lambda}_{l,l}^{(3)} + \sum_{l=1}^{d}\widetilde{\mathbf{E}}_{\boldsymbol{\pi}_l,\boldsymbol{\pi}_l}\left(\widetilde{\mathbf{\Lambda}}_{l,l}^{(3)} - \mathbf{\Lambda}_{l,l}^{(3)}\right)$$

$$= \sum_{l=1}^{d}\mathbf{E}_{\boldsymbol{\pi}_l,\boldsymbol{\pi}_l}\mathbf{\Lambda}_{l,l}^{(3)} + \operatorname{Trace}\left(\mathbf{\Pi}(\widetilde{\mathbf{E}}-\mathbf{E})\mathbf{\Pi}^{\top}\widetilde{\mathbf{\Lambda}}^{(3)}\right) + \operatorname{Trace}\left(\mathbf{\Pi}\widetilde{\mathbf{E}}\mathbf{\Pi}^{\top}(\widetilde{\mathbf{\Lambda}}^{(3)} - \mathbf{\Lambda}^{(3)})\right).$$

We apply Cauchy-Schwarz inequality again to the second and the third term:

$$\operatorname{Trace}\left(\mathbf{\Pi}(\widetilde{\mathbf{E}}-\mathbf{E})\mathbf{\Pi}^{\top}\widetilde{\mathbf{\Lambda}}^{(3)}\right) \leq \|(\widetilde{\mathbf{E}}-\mathbf{E})\mathbf{\Pi}^{\top}\|_{\mathrm{F}}\|\mathbf{\Pi}^{\top}\widetilde{\mathbf{\Lambda}}^{(3)}\|_{\mathrm{F}} = \|\widetilde{\mathbf{E}}-\mathbf{E}\|_{\mathrm{F}}\|\widetilde{\mathbf{\Lambda}}^{(3)}\|_{\mathrm{F}},$$

$$\operatorname{Trace}\left(\mathbf{\Pi}\widetilde{\mathbf{E}}\mathbf{\Pi}^{\top}(\widetilde{\mathbf{\Lambda}}^{(3)} - \mathbf{\Lambda}^{(3)})\right) \leq \|\widetilde{\mathbf{E}}\mathbf{\Pi}^{\top}\|_{\mathrm{F}}\|\mathbf{\Pi}^{\top}(\widetilde{\mathbf{\Lambda}}^{(3)} - \mathbf{\Lambda}^{(3)})\|_{\mathrm{F}} = \|\widetilde{\mathbf{E}}\|_{\mathrm{F}}\|\widetilde{\mathbf{\Lambda}}^{(3)} - \mathbf{\Lambda}^{(3)}\|_{\mathrm{F}}$$

where we use invariance of the Frobenius norm under column and row permutations. We conclude that

$$\operatorname{Trace}\left(\widetilde{\mathbf{E}}\mathbf{Q}^{**}\widetilde{\mathbf{N}}(\mathbf{Q}^{**})^{\top}\right) \leq \sum_{l=1}^{d}\mathbf{E}_{\boldsymbol{\pi}_l,\boldsymbol{\pi}_l}\mathbf{\Lambda}_{l,l}^{(3)} + \|\widetilde{\mathbf{E}}-\mathbf{E}\|_{\mathrm{F}}\|\widetilde{\mathbf{\Lambda}}^{(3)}\|_{\mathrm{F}} + \|\widetilde{\mathbf{E}}\|_{\mathrm{F}}\|\widetilde{\mathbf{\Lambda}}^{(3)} - \mathbf{\Lambda}^{(3)}\|_{\mathrm{F}}.$$

We combine this inequality with (49) and obtain:

$$\operatorname{Trace}\left(\mathbf{E}\mathbf{Q}^{**}\mathbf{N}(\mathbf{Q}^{**})^{\top}\right) \leq \sum_{l=1}^{d}\mathbf{E}_{\boldsymbol{\pi}_l,\boldsymbol{\pi}_l}\mathbf{\Lambda}_{l,l}^{(3)} + 2\|\mathbf{E}\|_{\mathrm{F}}\|\mathbf{\Lambda}^{(3)}-\widetilde{\mathbf{\Lambda}}^{(3)}\|_{\mathrm{F}} + 2\|\mathbf{E}-\widetilde{\mathbf{E}}\|_{\mathrm{F}}\|\widetilde{\mathbf{\Lambda}}^{(3)}\|_{\mathrm{F}}. \quad (50)$$

Next, we use triangle inequality and deduce that

$$\|\widetilde{\mathbf{\Lambda}}^{(3)}\|_{\mathrm{F}} \leq \|\mathbf{\Lambda}^{(3)}\|_{\mathrm{F}} + \|\widetilde{\mathbf{\Lambda}}^{(3)} - \mathbf{\Lambda}^{(3)}\|_{\mathrm{F}}.$$

Hence, we continue (50):

$$\operatorname{Trace}\left(\mathbf{E}\mathbf{Q}^{**}\mathbf{N}(\mathbf{Q}^{**})^{\top}\right) \leq \sum_{l=1}^{d}\mathbf{E}_{\boldsymbol{\pi}_l,\boldsymbol{\pi}_l}\mathbf{\Lambda}_{l,l}^{(3)} + 2\|\mathbf{E}\|_{\mathrm{F}}\|\mathbf{\Lambda}^{(3)}$$

$$-\widetilde{\boldsymbol{\Lambda}}^{(3)}\|_{\mathrm{F}} + 2\|\mathbf{E} - \widetilde{\mathbf{E}}\|_{\mathrm{F}}\left(\|\boldsymbol{\Lambda}^{(3)}\|_{\mathrm{F}} + \|\widetilde{\boldsymbol{\Lambda}}^{(3)} - \boldsymbol{\Lambda}^{(3)}\|_{\mathrm{F}}\right)$$

$$= \sum_{l=1}^{d} \mathbf{E}_{\boldsymbol{\pi}_l,\boldsymbol{\pi}_l}\boldsymbol{\Lambda}_{l,l}^{(3)} + 2\|\mathbf{E}\|_{\mathrm{F}}\|\boldsymbol{\Lambda}^{(3)} - \widetilde{\boldsymbol{\Lambda}}^{(3)}\|_{\mathrm{F}} + 2\|\boldsymbol{\Lambda}^{(3)}\|_{\mathrm{F}}\|\mathbf{E} - \widetilde{\mathbf{E}}\|_{\mathrm{F}} + 2\|\mathbf{E} - \widetilde{\mathbf{E}}\|_{\mathrm{F}}\|\widetilde{\boldsymbol{\Lambda}}^{(3)} - \boldsymbol{\Lambda}^{(3)}\|_{\mathrm{F}}$$

$$\leq \sum_{l=1}^{d} \mathbf{E}_{\boldsymbol{\pi}_l,\boldsymbol{\pi}_l}\boldsymbol{\Lambda}_{l,l}^{(3)} + 2\|\mathbf{E}\|_{\mathrm{F}}\|\boldsymbol{\Lambda}^{(3)} - \widetilde{\boldsymbol{\Lambda}}^{(3)}\|_{\mathrm{F}} + 2\|\boldsymbol{\Lambda}^{(3)}\|_{\mathrm{F}}\|\mathbf{E} - \widetilde{\mathbf{E}}\|_{\mathrm{F}} + 2\|\mathbf{E}\|_{\mathrm{F}}\|\widetilde{\boldsymbol{\Lambda}}^{(3)} - \boldsymbol{\Lambda}^{(3)}\|_{\mathrm{F}}$$

where in the last transition we use $\|\mathbf{E} - \widetilde{\mathbf{E}}\|_{\mathrm{F}} \leq \|\mathbf{E}\|_{\mathrm{F}}$ which is according to (47). We continue this chain of inequalities using (47) again:

$$\mathrm{Trace}\left(\mathbf{E}\mathbf{Q}^{**}\mathbf{N}(\mathbf{Q}^{**})^{\top}\right) \leq \sum_{l=1}^{d} \mathbf{E}_{\boldsymbol{\pi}_l,\boldsymbol{\pi}_l}\boldsymbol{\Lambda}_{l,l}^{(3)} + \frac{2}{12}\delta + \frac{2}{12}\delta + \frac{2}{12}\delta = \sum_{l=1}^{d} \mathbf{E}_{\boldsymbol{\pi}_l,\boldsymbol{\pi}_l}\boldsymbol{\Lambda}_{l,l}^{(3)} + \frac{\delta}{2}$$

$$< \sum_{l=1}^{d} \mathbf{E}_{\boldsymbol{\pi}_l,\boldsymbol{\pi}_l}\boldsymbol{\Lambda}_{l,l}^{(3)} + \delta.$$

This is a contradiction with (46) taking into account $\mathbf{Q}^{**}$'s definition (48). Hence, (44) is proven. $\square$

Let $\mathbf{Q}^*$ be defined as in Lemma A.4's statement. Further, we denote $\boldsymbol{\pi}(\mathbf{A}) = \boldsymbol{\pi}$, $\boldsymbol{\Pi}(\mathbf{A}) = \boldsymbol{\Pi}$ where $\boldsymbol{\pi}, \boldsymbol{\Pi}$ are defined as in Lemma A.4's statement. That is, $\boldsymbol{\pi}(\mathbf{A})$ denotes some permutation which sorts diagonal entries of $\mathbf{E}$ in a non-ascending order. It's a function of $\mathbf{A}$ since $\mathbf{E}$ is a function of $\mathbf{A}$ defined in (35). In fact, based on (35), we see that $\boldsymbol{\pi}(\mathbf{A})$ is some permutation which sorts diagonal entries of $\mathbf{A}$ in a non-descending order. $\boldsymbol{\Pi}(\mathbf{A})$ denotes a permutation matrix corresponding to $\boldsymbol{\pi}(\mathbf{A})$. That is, diagonal entries of $\boldsymbol{\Pi}(\mathbf{A})\mathbf{A}\boldsymbol{\Pi}(\mathbf{A})^{\top}$ are sorted in a non-descending order.

Let $\mathcal{G}(\mathbf{A})$ denote the right hand side of (37) where we substitute $\mathbf{Q} = \mathbf{Q}^*$. That is, $\mathcal{G}(\mathbf{A})$ is an optimal value of $\overline{\mathcal{L}}(\boldsymbol{\theta}_{\mathrm{SDE}}; \mathcal{X}, \mathcal{Y}, \mathcal{T}_{\mathrm{SDE}})$ with $\mathbf{A}$ fixed:

$$\mathcal{G}(\mathbf{A}) = \log\det(\mathbf{I}_d - 4\mathbf{A}) - \frac{1}{2}\log\det(\mathbf{I}_d - 8\mathbf{A}) - L^{-1}\sum_{i=1}^{L}\|\mathbf{x}^{(i)}\|^2 - L^{-1}\sum_{j=1}^{L}\|\mathbf{y}^{(j)}\|^2$$

$$- 2\sum_{l=1}^{d} \mathbf{E}_{\boldsymbol{\pi}(\mathbf{A})_l,\boldsymbol{\pi}(\mathbf{A})_l}\boldsymbol{\Lambda}_{l,l}^{(3)}$$

$$= \log\det(\mathbf{I}_d - 4\mathbf{A}) - \frac{1}{2}\log\det(\mathbf{I}_d - 8\mathbf{A}) - L^{-1}\sum_{i=1}^{L}\|\mathbf{x}^{(i)}\|^2 - L^{-1}\sum_{j=1}^{L}\|\mathbf{y}^{(j)}\|^2$$

$$+ \sum_{l=1}^{d}\left(1 + (1 - 8\mathbf{A}_{\boldsymbol{\pi}(\mathbf{A})_l,\boldsymbol{\pi}(\mathbf{A})_l})^{-1}\right)\boldsymbol{\Lambda}_{l,l}^{(3)} \tag{51}$$

where we use $\mathbf{E}$'s definition (35). Let $\boldsymbol{\pi}^{-1}(\mathbf{A}) \in \mathbb{N}^d$ denote a permutation inverse to $\boldsymbol{\pi}(\mathbf{A})$. By rearranging terms in the sum, we have:

$$\sum_{l=1}^{d}\left(1 + (1 - 8\mathbf{A}_{\boldsymbol{\pi}(\mathbf{A})_l,\boldsymbol{\pi}(\mathbf{A})_l})^{-1}\right)\boldsymbol{\Lambda}_{l,l}^{(3)} = \sum_{l=1}^{d}\left(1 + (1 - 8\mathbf{A}_{l,l})^{-1}\right)\boldsymbol{\Lambda}_{\boldsymbol{\pi}^{-1}(\mathbf{A})_l,\boldsymbol{\pi}^{-1}(\mathbf{A})_l}^{(3)}.$$

Therefore, we have:

$$\mathcal{G}(\mathbf{A}) = \log\det(\mathbf{I}_d - 4\mathbf{A}) - \frac{1}{2}\log\det(\mathbf{I}_d - 8\mathbf{A}) - L^{-1}\sum_{i=1}^{L}\|\mathbf{x}^{(i)}\|^2 - L^{-1}\sum_{j=1}^{L}\|\mathbf{y}^{(j)}\|^2$$

$$+ \sum_{l=1}^{d}\left(1 + (1 - 8\mathbf{A}_{l,l})^{-1}\right)\boldsymbol{\Lambda}_{\boldsymbol{\pi}^{-1}(\mathbf{A})_l,\boldsymbol{\pi}^{-1}(\mathbf{A})_l}^{(3)}. \tag{52}$$

Define a new function $\mathcal{G}(\widehat{\mathbf{A}}, \mathbf{A})$, where $\widehat{\mathbf{A}} \in \mathbb{D}_d$ satisfies $8\widehat{\mathbf{A}} \prec \mathbf{I}_d$, as follows:

$$\mathcal{G}(\widehat{\mathbf{A}}, \mathbf{A}) = \log \det(\mathbf{I}_d - 4\widehat{\mathbf{A}}) - \frac{1}{2} \log \det(\mathbf{I}_d - 8\widehat{\mathbf{A}}) + \sum_{l=1}^{d}(1 - 8\widehat{\mathbf{A}}_{l,l})^{-1} \mathbf{\Lambda}^{(3)}_{\boldsymbol{\pi}^{-1}(\mathbf{A})_l, \boldsymbol{\pi}^{-1}(\mathbf{A})_l}.$$

By the definition of $\mathcal{G}(\widehat{\mathbf{A}}, \mathbf{A})$, we have:

$$\mathcal{G}(\mathbf{A}) = \mathcal{G}(\mathbf{A}, \mathbf{A}) - L^{-1}\sum_{i=1}^{L} \|\mathbf{x}^{(i)}\|^2 - L^{-1}\sum_{j=1}^{L} \|\mathbf{y}^{(j)}\|^2 + \sum_{l=1}^{d} \mathbf{\Lambda}^{(3)}_{\boldsymbol{\pi}^{-1}(\mathbf{A})_l, \boldsymbol{\pi}^{-1}(\mathbf{A})_l}.$$

Hence, it holds:

$$\mathcal{G}(\mathbf{A}) \geq \inf_{\widehat{\mathbf{A}} \in \mathbb{D}_d, \, 8\widehat{\mathbf{A}} \prec \mathbf{I}_d} \mathcal{G}(\widehat{\mathbf{A}}, \mathbf{A}) - L^{-1}\sum_{i=1}^{L} \|\mathbf{x}^{(i)}\|^2 - L^{-1}\sum_{j=1}^{L} \|\mathbf{y}^{(j)}\|^2 + \sum_{l=1}^{d} \mathbf{\Lambda}^{(3)}_{\boldsymbol{\pi}^{-1}(\mathbf{A})_l, \boldsymbol{\pi}^{-1}(\mathbf{A})_l}. \quad (53)$$

Next, we show that there is a closed-form expression for the solution of $\inf_{\widehat{\mathbf{A}} \in \mathbb{D}_d, \, 8\widehat{\mathbf{A}} \prec \mathbf{I}_d} \mathcal{G}(\widehat{\mathbf{A}}, \mathbf{A})$. Since $\widehat{\mathbf{A}} \in \mathbb{D}_d$, we have: $\log \det(\mathbf{I}_d - 4\widehat{\mathbf{A}}) = \sum_{l=1}^{d} \log(1 - 4\widehat{\mathbf{A}}_{l,l})$, $\log \det(\mathbf{I}_d - 8\widehat{\mathbf{A}}) = \sum_{l=1}^{d} \log(1 - 8\widehat{\mathbf{A}}_{l,l})$. We further have:

$$\mathcal{G}(\widehat{\mathbf{A}}, \mathbf{A}) = \sum_{l=1}^{d} \left( \log(1 - 4\widehat{\mathbf{A}}_{l,l}) - \frac{1}{2} \log(1 - 8\widehat{\mathbf{A}}_{l,l}) + (1 - 8\widehat{\mathbf{A}}_{l,l})^{-1} \mathbf{\Lambda}^{(3)}_{\boldsymbol{\pi}^{-1}(\mathbf{A})_l, \boldsymbol{\pi}^{-1}(\mathbf{A})_l} \right). \quad (54)$$

From (54), we see that minimization $\inf_{\widehat{\mathbf{A}} \in \mathbb{D}_d, \, 8\widehat{\mathbf{A}} \prec \mathbf{I}_d} \mathcal{G}(\widehat{\mathbf{A}}, \mathbf{A})$ reduces to $d$ independent minimization problems with respect to $\widehat{\mathbf{A}}_{l,l}$ such that $8\widehat{\mathbf{A}}_{l,l} < 1$. $l$'th problem, $1 \leq l \leq d$, is solved using Lemma A.1 where we set $\phi = \mathbf{\Lambda}^{(3)}_{\boldsymbol{\pi}^{-1}(\mathbf{A})_l, \boldsymbol{\pi}^{-1}(\mathbf{A})_l}$. Let $\mathbf{A}^{**} \in \mathbb{D}_d$ denote the corresponding solution. Then, for all $1 \leq l \leq d$, we have:

$$\mathbf{A}^{**}_{l,l} = \frac{1}{16} \left( 1 - 2\mathbf{\Lambda}^{(3)}_{\boldsymbol{\pi}^{-1}(\mathbf{A})_l, \boldsymbol{\pi}^{-1}(\mathbf{A})_l} - \sqrt{\left( 2\mathbf{\Lambda}^{(3)}_{\boldsymbol{\pi}^{-1}(\mathbf{A})_l, \boldsymbol{\pi}^{-1}(\mathbf{A})_l} + 1 \right)^2 + 8\mathbf{\Lambda}^{(3)}_{\boldsymbol{\pi}^{-1}(\mathbf{A})_l, \boldsymbol{\pi}^{-1}(\mathbf{A})_l}} \right). \quad (55)$$

From (53) it follows that

$$\mathcal{G}(\mathbf{A}) \geq \mathcal{G}(\mathbf{A}^{**}, \mathbf{A}) - L^{-1}\sum_{i=1}^{L} \|\mathbf{x}^{(i)}\|^2 - L^{-1}\sum_{j=1}^{L} \|\mathbf{y}^{(j)}\|^2 + \sum_{l=1}^{d} \mathbf{\Lambda}^{(3)}_{\boldsymbol{\pi}^{-1}(\mathbf{A})_l, \boldsymbol{\pi}^{-1}(\mathbf{A})_l}$$

$$= \log \det(\mathbf{I}_d - 4\mathbf{A}^{**}) - \frac{1}{2} \log \det(\mathbf{I}_d - 8\mathbf{A}^{**}) - L^{-1}\sum_{i=1}^{L} \|\mathbf{x}^{(i)}\|^2 - L^{-1}\sum_{j=1}^{L} \|\mathbf{y}^{(j)}\|^2$$

$$+ \sum_{l=1}^{d} \left(1 + (1 - 8\mathbf{A}^{**}_{l,l})^{-1}\right) \mathbf{\Lambda}^{(3)}_{\boldsymbol{\pi}^{-1}(\mathbf{A})_l, \boldsymbol{\pi}^{-1}(\mathbf{A})_l}. \quad (56)$$

Denote $\mathbf{E}^{**} = -\frac{1}{2}\mathbf{I}_d - \frac{1}{2}(\mathbf{I}_d - 8\mathbf{A}^{**})^{-1}$. Then we have:

$$\sum_{l=1}^{d} \left(1 + (1 - 8\mathbf{A}^{**}_{l,l})^{-1}\right) \mathbf{\Lambda}^{(3)}_{\boldsymbol{\pi}^{-1}(\mathbf{A})_l, \boldsymbol{\pi}^{-1}(\mathbf{A})_l} = -2\text{Trace}\left( \mathbf{E}^{**} \mathbf{\Pi}(\mathbf{A})^{-1} \mathbf{\Lambda}^{(3)} \left( \mathbf{\Pi}(\mathbf{A})^{-1} \right)^{\top} \right)$$

$$\leq -2\sum_{l=1}^{d} \mathbf{E}^{**}_{\boldsymbol{\pi}(\mathbf{A}^{**})_l, \boldsymbol{\pi}(\mathbf{A}^{**})_l} \mathbf{\Lambda}^{(3)}_{l,l} \quad (57)$$

where the second transition follows from Lemma A.4 and the fact that $\boldsymbol{\pi}(\mathbf{A}^{**})$ sorts diagonal entries of $\mathbf{A}^{**}$ in a non-descending order, hence its sorts diagonal entries of $\mathbf{E}^{**}$ in a non-ascending order (recall the definition of $\boldsymbol{\pi}(\mathbf{A}^{**})$ and $\mathbf{E}^{**}$). Denote $\mathbf{E}^{*} = \mathbf{\Pi}(\mathbf{A}^{**})\mathbf{E}^{**}\mathbf{\Pi}(\mathbf{A}^{**})^{\top}$. Then $\mathbf{E}^{**}_{\boldsymbol{\pi}(\mathbf{A}^{**})_l, \boldsymbol{\pi}(\mathbf{A}^{**})_l} = \mathbf{E}^{*}_{l,l}$ for all $1 \leq l \leq d$ and

$$\sum_{l=1}^{d} \mathbf{E}^{**}_{\boldsymbol{\pi}(\mathbf{A}^{**})_l, \boldsymbol{\pi}(\mathbf{A}^{**})_l} \mathbf{\Lambda}^{(3)}_{l,l} = \sum_{l=1}^{d} \mathbf{E}^{*}_{l,l} \mathbf{\Lambda}^{(3)}_{l,l}. \quad (58)$$

Further, we have:

$$\mathbf{E}^* = \mathbf{\Pi}(\mathbf{A}^{**})\left(-\frac{1}{2}\mathbf{I}_d - \frac{1}{2}(\mathbf{I}_d - 8\mathbf{A}^{**})^{-1}\right)\mathbf{\Pi}(\mathbf{A}^{**})^\top$$

$$= -\frac{1}{2}\mathbf{I}_d - \frac{1}{2}(\mathbf{I}_d - 8\mathbf{\Pi}(\mathbf{A}^{**})\mathbf{A}^{**}\mathbf{\Pi}(\mathbf{A}^{**})^\top)^{-1}$$

$$= -\frac{1}{2}\mathbf{I}_d - \frac{1}{2}(\mathbf{I}_d - 8\mathbf{A}^*)^{-1}$$

where we denote $\mathbf{A}^* = \mathbf{\Pi}(\mathbf{A}^{**})\mathbf{A}^{**}\mathbf{\Pi}(\mathbf{A}^{**})^\top$, i.e. $\mathbf{A}^*_{l,l} = \mathbf{A}^{**}_{\boldsymbol{\pi}(\mathbf{A}^{**})_l,\boldsymbol{\pi}(\mathbf{A}^{**})_l}$ for all $1 \leq l \leq d$. Given the definition of $\mathbf{A}^{**}$ (55), for all $1 \leq l \leq d$ we have:

$$\mathbf{A}^*_{l,l} = \frac{1}{16}\left(1 - 2\mathbf{\Lambda}^{(3)}_{l,l} - \sqrt{\left(2\mathbf{\Lambda}^{(3)}_{l,l} + 1\right)^2 + 8\mathbf{\Lambda}^{(3)}_{l,l}}\right). \tag{59}$$

That is, $\mathbf{A}^*$ is independent of $\mathbf{A}$. Based on (59), we see that smaller values of $\mathbf{\Lambda}^{(3)}_{l,l}$ result in bigger values of $\mathbf{A}^*_{l,l}$. Since $\mathbf{\Lambda}^{(3)}_{1,1},\dots,\mathbf{\Lambda}^{(3)}_{d,d}$ are ordered in a non-ascending order, we deduce that $\mathbf{A}^*_{1,1},\dots,\mathbf{A}^*_{d,d}$ are ordered in a non-descending order. By the definition of $\boldsymbol{\pi}(\mathbf{A}^*)$, we then have $\mathbf{A}^*_{l,l} = \mathbf{A}^*_{\boldsymbol{\pi}(\mathbf{A}^*)_l,\boldsymbol{\pi}(\mathbf{A}^*)_l}$ for all $1 \leq l \leq d$. Therefore,

$$\sum_{l=1}^{d}\mathbf{E}^*_{l,l}\mathbf{\Lambda}^{(3)}_{l,l} = -\frac{1}{2}\sum_{l=1}^{d}(1 + (1 - 8\mathbf{A}^*_{l,l})^{-1})\mathbf{\Lambda}^{(3)}_{l,l} = -\frac{1}{2}\sum_{l=1}^{d}(1 + (1 - 8\mathbf{A}^*_{\boldsymbol{\pi}(\mathbf{A}^*)_l,\boldsymbol{\pi}(\mathbf{A}^*)_l})^{-1})\mathbf{\Lambda}^{(3)}_{l,l}.$$

Combining this with (58), (57), we can continue the chain of inequalities (56):

$$\mathcal{G}(\mathbf{A}) \geq \log\det(\mathbf{I}_d - 4\mathbf{A}^{**}) - \frac{1}{2}\log\det(\mathbf{I}_d - 8\mathbf{A}^{**}) - L^{-1}\sum_{i=1}^{L}\|\mathbf{x}^{(i)}\|^2 - L^{-1}\sum_{j=1}^{L}\|\mathbf{y}^{(j)}\|^2$$

$$+ \sum_{l=1}^{d}\left(1 + (1 - 8\mathbf{A}^*_{\boldsymbol{\pi}(\mathbf{A}^*)_l,\boldsymbol{\pi}(\mathbf{A}^*)_l})^{-1}\right)\mathbf{\Lambda}^{(3)}_{l,l}$$

$$= \log\det(\mathbf{I}_d - 4\mathbf{A}^*) - \frac{1}{2}\log\det(\mathbf{I}_d - 8\mathbf{A}*) - L^{-1}\sum_{i=1}^{L}\|\mathbf{x}^{(i)}\|^2 - L^{-1}\sum_{j=1}^{L}\|\mathbf{y}^{(j)}\|^2$$

$$+ \sum_{l=1}^{d}\left(1 + (1 - 8\mathbf{A}^*_{\boldsymbol{\pi}(\mathbf{A}^*)_l,\boldsymbol{\pi}(\mathbf{A}^*)_l})^{-1}\right)\mathbf{\Lambda}^{(3)}_{l,l} = \mathcal{G}(\mathbf{A}^*) \tag{60}$$

where in the second transition we use the fact that

$$\det(\mathbf{I}_d - 4\mathbf{A}^*) = \det\left(\mathbf{\Pi}(\mathbf{A}^{**})(\mathbf{I}_d - 4\mathbf{A}^{**})\mathbf{\Pi}(\mathbf{A}^{**})^\top\right) = \det\left(\mathbf{I}_d - 4\mathbf{A}^{**}\right)$$

and, similarly, $\det(\mathbf{I}_d - 8\mathbf{A}^*) = \det(\mathbf{I}_d - 8\mathbf{A}^{**})$. In the third transition, we use definition of $\mathcal{G}(\cdot)$ (51). Note that (60) holds for all $\mathbf{A} \in \mathbb{D}_d$ such that $8\mathbf{A} \prec \mathbf{I}_d$ and also $8\mathbf{A}^* \prec \mathbf{I}_d$ since $8\mathbf{A}^{**} \prec \mathbf{I}_d$. We conclude that, when $\mathbf{B}, \mathbf{C}, D$ are chosen optimally with a given $\mathbf{A}$, the minimum of $\overline{\mathcal{L}}(\boldsymbol{\theta}_{\mathrm{SDE}};\mathcal{X},\mathcal{Y},\mathcal{T}_{\mathrm{SDE}})$ is reached when $\mathbf{A} = \mathbf{A}^*$. As we have already deduced, diagonal entries of $\mathbf{A} = \mathbf{A}^*$ are sorted in the non-descending order. Hence, using Lemma A.4's notation, diagonal entries of $\mathbf{E}$ are already sorted in a non-ascending sorting order and $\boldsymbol{\pi} = (1,\dots,d)$, $\mathbf{\Pi} = \mathbf{I}_d$ satisfy requirements of the Lemma. Hence, with $\mathbf{A} = \mathbf{A}^*$, the optimal $\mathbf{B}$ has a form $(\mathbf{I}_d - 4\mathbf{A})^{1/2}\mathbf{Q}^*$ where $\mathbf{Q}^* = \mathbf{I}_d(\mathbf{Q}^{(3)})^\top = (\mathbf{Q}^{(3)})^\top$. Optimal $\mathbf{C}$ and $D$ are further determined by (33). (11) follows from (60) and the fact that, as discussed above, we can replace $\boldsymbol{\pi}(\mathbf{A}^*)_l$ with $l$ in (60), $1 \leq l \leq d$. $\qquad\square$

## B   Additional experimental details

### B.1   Compute resources and implementation

We use NumPy [25] in Google Colaboratory the variance comparison and kernel classification experiment. For the Transformer setups, we use TPU cluster and JAX [7] library. All tested Transformer variants were trained and tested on a TPU pods containing 4 TPU v3 chips with JAX and on GPUs (V100).

## B.2 Variance comparison

We repeat the setup of [31] closely: we draw 5 pairs of sets $\{\mathbf{x}^{(i)}\}_{1 \leq i \leq L}$, $\{\mathbf{y}^{(j)}\}_{1 \leq j \leq L}$, $L = 1024$. On each pair, we compute the relative variance for all pairs of points and for all indicated RF methods. Further, the shifted log-variance is optimized separately on each pair of sets for GERF, ADERF and SDERF.

We take $M = 1$ since $M$'s value is not important in this experiment: bigger $M$ would just shift the curves below. The reported curves are means over all pairs of points and over all 5 sets.

Table 2: Hyperparameters for the base models for pre-training for all methods

| Parameter | Value |
|---|---|
| # of heads | 12 |
| # of hidden layers | 12 |
| Hidden layer size | 768 |
| # of tokens | 512 |
| Batch size | 256 |
| M | 256 |
| Pretrain Steps | $1M$ |
| Loss | MLM |
| Activation layer | gelu |
| Dropout prob | 0.1 |
| Attention dropout prob | 0.1 |
| Optimizer | Adam |
| Learning rate | $10^{-4}$ |
| Compute resources | $8 \times 8$ TPUv3 |

Table 3: Dataset used for pre training.

| Dataset | # tokens | Avg. doc len. |
|---|---|---|
| Books [54] | 1.0B | 37K |
| Wikipedia | 3.1B | 592 |

## B.3 Kernel classification

As in [31], we obtain training, validation and test splits by shuffling the raw dataset and taking $90\%$, $5\%$, $5\%$ objects respectively. The splits are fixed for all RF methods. We tune $\sigma$ on a logarithmic grid of 10 values on $[10^{-2}, 10^2]$. For each $\sigma$ and each RF type, we try 50 seeds for drawing RFs during validation and testing. Testing is performed for the best $\sigma$ only. Figure 3 reports averages over 50 seeds. We use orthogonal $\omega$'s for all types of RFs as described in [31], since orthogonal random features work better in practice [16, 31].

## B.4 DERFs for long-sequence Transformers

### B.4.1 Speech modelling

Our Conformer-Transducer variant was characterized by: **20** conformer layers, model_dim = **512**, relative position embedding dimensionality rped = **512** and $h = \mathbf{8}$ heads. We used batch size bs = **2048** and trained with the adam optimizer on TPUs. For the regular Conformer-Transducer training, we run ablation studies over different number of random features: $m = \mathbf{8}, \mathbf{32}, \mathbf{128}$. In the NST setting, we run experiments with $m = \mathbf{8}$. We reported commonly used metric: normalized word error rate (NWER).

### B.4.2 Natural language processing

We pretrained BERT model on two publicly available datasets (see: Table 3). Following the original BERT training, we mask $15\%$ of tokens in these two datasets, and train to predict the mask. All methods were warm started from exactly the same pre-trained checkpoint after 1M iteration of BERT

pretraining. We used the exact same hyperparameter-setup for all the baselines (FAVOR++[31], FAVOR+ [16], ELU [26], ReLU [16]) and FAVOR++. The hyperparameters for pretraining are shown in Table 2. We finetuned on GLUE task, warm-starting with the weights of the pretrained model. The setup is analogous to the one from the original BERT paper.

