# OpenReview forum: "Dense-Exponential Random Features: Sharp Positive Estimators of the Gaussian Kernel"
_NeurIPS.cc/2023/Conference — NeurIPS 2023 poster_

### Official Review · Reviewer_z18t · 2023-07-05

**Soundness:** 4 excellent
**Presentation:** 3 good
**Contribution:** 2 fair
**Rating:** 3
**Confidence:** 4

**Summary:**

This paper studies the problem of computing the matrix "KC". Here, K is the kernel matrix (L-by-L, with L very large, could be reproducing kernel, but OK otherwise), and C is a known constant matrix. The kernel is restricted to be "scaled softmax kernel": exp(x-dot-y)F(x)F(y) (for some function F not difficult to handle). The contribution of this paper, is the design of a class of random features, of which the correlation is exactly K, but to some extent, the variance of replacing the mean (for the correlation) by sample mean, can be minimized with closed form. This research, to the best of my knowledge, is innovative, inspiring, and interesting.

**Strengths:**

In this paper, the idea, and the derived formulation of optimizing sample variance for improved kernel estimation is innovative. Besides the notation problem raised below, the paper has a clear and friendly structure that is easy to follow. The main results are presented clearly. These suggest excellent writing quality and clarity. We do have concerns on significance, but we would like to raise such concerns below, instead of here.

**Weaknesses:**

1. Concerns about the writing. I feel that the notations are unnecessarily too heavy. For example, I can not see any reason that blocks one from rewriting:
f^(k) --> f^k (or just f and g), B^(k) --> B_k, C^(k) --> C_k or C^k, x^(i) --> x_i, y^(i) --> y_i, M^(k), Q^(k), Lambda^(k), .... By the rewriting, the number of notations is reduced by 50%! Some notations only appear a few times and can just be replaced by English, e.g., O_d (the set of orthogonal matrices).
2. The significance of the paper relies on the significance of the exponential kernel exp(x-dot-y). If computing the transformer attention is the only application, the vanilla computational cost should be compared. Otherwise, I feel not certain if this is a widely used kernel family, and (e.g., for Gaussian kernel) in the application scenarios (e.g., regularized least squares, or support vector machines), whether computing KC is really the bottleneck.

**Questions:**

1. Lines 76--77. Could you please provide more details on why the computational complexity of hat-K-times-C is O(LMn)?
2. In Eq. (8), the variable "theta" does not show up in the right-hand side, and is not elsewhere defined. Better if this variable can be defined.
3. It remains very unclear even after a careful study of Section 3, what are "homogeneity heuristic" and "a certain optimization problem" mentioned in Lines 124 and 125. Better if these can be explicitly pointed out.
4. Consider the mission of computing the matrix KC, as elaborated in Section 2.1, whether adopting DERFs would reduce computational complexity? To be specific, whether the amount of variance reduced, deserves the computational cost incurred by optimizing the random features via the approaches introduced in Section 4? Can one achieve the same accuracy and computation complexity simply by using a larger M? For example, Theorem 4.2.

**Limitations:**

foundational research, limitations well managed.

---

> ### Author Rebuttal · Authors · 2023-08-08
>
> We would like to sincerely thank the Reviewer for all the comments. We address weaknesses and questions below.
>
> > Concerns about the writing.
>
> We will address this concern by simplifying and de-densifying notation in the revision of the paper.
>
> > Otherwise, I feel not certain if this is a widely used kernel family, and … whether computing KC is really the bottleneck.
>
> Note that our derivations, as mentioned in the title and in e.g. lines 57-58, apply for both softmax and Gaussian kernels. Gaussian kernel is a very important object of research in the kernel methods literature and a lot of efforts have been made to speed it up in situations where the kernel matrix is big (e.g. in SVM or kernel regression where the number of data points is above a few thousands) – see [38,39,40] and thousands of follow ups of these papers.
>
> > Lines 76--77. Could you please provide more details on why the computational complexity of hat-K-times-C is O(LMn)?
>
> As mentioned in the text, hat{K} = P S^T. Hence, hat{K} C = P S^T C = P (S^T C). The complexity of computing U = S^T C is O(LMn) since S^T, C are of shapes MxL and Lxn. The complexity of computing P U is also O(LMn) since P is of shape LxM and U is of shape Mxn.
>
> > In Eq. (8), the variable "theta" does not show up in the right-hand side, and is not elsewhere defined. Better if this variable can be defined.
>
> Theta is defined in line 126. We will expand its definition to make it clearer in the final revision.
>
> > It remains very unclear even after a careful study of Section 3, what are "homogeneity heuristic" and "a certain optimization problem" mentioned in Lines 124 and 125.
>
> “homogeneity heuristic” is defined in line 116 and the optimization problem is the minimization of (8) as mentioned in line 129. We will clarify these notations in the final version.
>
> > Can one achieve the same accuracy and computation complexity simply by using a larger M?
>
> As we demonstrate in our experimental results (Figure 1), our RF variants result in up to e^10 variance improvements over the previous best variant. To achieve that by using a larger M means taking e^20 * M random features (since variance is proportional to M^{-1/2}) which would be prohibitively expensive.

---

> > ### Comment · Reviewer_z18t · 2023-08-16
> > **Thank you for the feedback**
> >
> > All of my concerns are resolved. I would like to raise the rating after the next phase of discussion with other reviewers.

---

### Official Review · Reviewer_PE4H · 2023-07-05

**Soundness:** 3 good
**Presentation:** 3 good
**Contribution:** 3 good
**Rating:** 7
**Confidence:** 3

**Summary:**

In this paper, the GERFs are genearilzied to dense exponential random features (DERFs). The paper shows that GERFs and PosRFs are the specific situation of DERFs, from which if follows that with a suitable parameter estimator, DERFs can have better performance.

**Strengths:**

As far as I know, the DERFs are novel and are a good extension from GERFs.

There are sufficient discussions and experiments from many aspects, all showing the good performance of the proposed method.

The paper is well organized and clearly represented.

**Weaknesses:**

I did not find significant weakness. One possible is that because this paper include many aspects, it may make the reader lose the focus.

It is better to clearly express the suggestion: in which case DERFs is recommended.


**Questions:**

In different experiments, several random features are compared when the number of random features are the same. How about the real calculation time? Since there could be difference on calculating different random features in the inference.

---

> ### Author Rebuttal · Authors · 2023-08-08
>
> We would like to thank the Reviewer for a high score and kind words in “Strengths”! We address weaknesses and questions below.
>
> > It is better to clearly express the suggestion: in which case DERFs is recommended.
>
> We recommend using them whenever the sequence length is too prohibitive for using exact kernel matrix/self-attention.
>
> > How about the real calculation time? Since there could be difference on calculating different random features in the inference.
>
> The computational complexity for all RF methods scales linearly with the sequence length, hence, for a large sequence length, we consider all these methods as efficient and only compare the downstream metric performance.

---

> > ### Comment · Reviewer_PE4H · 2023-08-18
> >
> > Thanks for the response. I am also glad to see some other reviewers' concerns have been solved. Overall, I keep my positive score.

---

### Official Review · Reviewer_LgLu · 2023-07-06

**Soundness:** 3 good
**Presentation:** 3 good
**Contribution:** 3 good
**Rating:** 6
**Confidence:** 2

**Summary:**

The authors propose new random features for Gaussian and softmax kernels, and apply the approximation to learning scalable Transformer networks.

**Strengths:**

1. The proposed dense exponential random features (DERFs) generalize the current positive random features (PosRFs) and  generalized exponential random features (GERFs).
2. The authors show stronger theoretical results for the proposed DERFs in scalable Transformer networks where the self-attention matrix is approximated as a low-rank matrix when the sequence is long.
3. The utility of the method is demonstrated on a variety of datasets with competitive results.


**Weaknesses:**

1. While the authors argue that significant variance reduction can be achieved, it is not clear whether the proposed approximation is unbiased.
2. The method could be computationally expensive---the term D involves computing the determinant.
3. Theorem 4.1 makes a few assumptions, restrictions of these conditions are not fully discussed.


**Questions:**

How feasible are the conditions in Theorem 4.1?

---

> ### Author Rebuttal · Authors · 2023-08-08
>
> Thank you for the review! We address weaknesses and questions below.
>
> > While the authors argue that significant variance reduction can be achieved, it is not clear whether the proposed approximation is unbiased.
>
> The proposed approximation is precisely unbiased according to Theorem 4.1.
>
> > The method could be computationally expensive---the term D involves computing the determinant.
>
> Note that A is a dxd diagonal matrix in all our DERF variants, hence computing its determinant takes O(d) time.
>
> > Theorem 4.1 makes a few assumptions, restrictions of these conditions are not fully discussed.
>
> All our close-form optimal solutions (Theorems 4.2, 4.3) satisfy constraints from Theorem 4.1 as mentioned in lines 176-177 for Theorem 4.2 and 198-200 for Theorem 4.1.
>
> > How feasible are the conditions in Theorem 4.1?
>
> As we mentioned above, these conditions are satisfied automatically for our optimal solutions.

---

> > ### Comment · Reviewer_LgLu · 2023-08-18
> > **After rebuttal**
> >
> > Thank you for addressing my comments. I've read all the reviews and my score remains unchanged.

---

### Official Review · Reviewer_pnU2 · 2023-07-12

**Soundness:** 3 good
**Presentation:** 4 excellent
**Contribution:** 3 good
**Rating:** 6
**Confidence:** 4

**Summary:**

This paper is the next instalment in a series of works focusing on low-rank transforners, e.g. [15, 30]. They propose Dense Exponential Random Features (DERF) for unbiased Monte Carlo approximation of Gaussian or softmax kernels. This class of features generalizes GERFs from previous work, with the main difference being that the features are parametrized by matrix-valued parameters rather than scalars. Hence, this flexibility allows for capturing a larger class of features for kernel approximation.  The parameters themselves can be analytically optimized in several useful special cases: ADERF, SDERF and SADERF. The parameter optimization is carried out with respect to a so-called shifted log-variance objective averaged over the dataset, and it is shown that optimizing this objective formalizes a "homogeneity heuristic" from previous work, which is a nice observation. In particular, this allows to select the same set of parameters across all data points. The experiments are essentially the same as in previous works: variance analysis on synthetic datasets + cifar + mnist, a set of nonparametric kernel regression benchmarks, speech modelling and low-rank uptraining on NLP tasks. Improvements compared to previous work and scalable transformers are achieved.

**Strengths:**

The paper is overall well written, it does really well at communicating the main issues and ideas on which it is based. There is both a theoretical and experimental component, both of which have some interesting results. The main idea is novel, and the proposed approach for generalizing previous random feature methods is non-trivial; I was impressed that the analytical calculations could also be carried out in the matrix-valued case. I believe it is of interest to both the kernel and efficient transformer communities to have some scalable and expressive random features, and the paper definitely takes another step in this direction.

**Weaknesses:**

Overall there are not that many weaknesses in my opinion. The impact is slightly less significant than of its predecessor [30], where the main novelty was allowing for low-rank uptraining of pre-trained Transformers, which was already achieved by FAVOR++ (efficient Transformer based on GERFs). Nevertheless, the improved experimental performance should be of interest.

I was missing some more comparisons between the variations: for one, SADERFs were not included in any of them, and only used for the last NLP task. Computationally, this variant seems to be the most favourable since it does not require the eigendecomposition of a $d \times d$ matrix, hence foregoing an $O(d^3)$ computational cost, which is associated with ADERF and SDERF. It would be interesting to see this variation included in the comparisons to have an idea about the trade-off between the performance and this computational saving.

I was also missing some intuition or hypothesis regarding why SDERF seems to perform best. Is it because it allows for a non-isotropic matrix $A$, hence allowing to adapt the variance of $\omega$ on a coordinate-by-coordinate basis in the quadratic form containing $\omega$, if this makes sense? It would be interesting to know which additional parameters result in the biggest improvement compared to GERF, so that focus could be placed on optimizing the scalability-performance trade-off. (See questions for more)

Another question that seems unaddressed to me is how much information is lost compared to a full Transformer on truly long-range tasks. At the moment, we do not know the "price" we pay for the subquadratic scalability in sequence length, and it would also be interesting to include some more long-range tasks, which compares full Transformer vs RF methods.

**Questions:**

Some questions and remarks:
- Is $\sigma$ analogous to the bandwidth parameter of the kernel in Figure 1 (right) and Figure 2? This is not stated explicitly.
- In line 138-139, it is stated that [30] achieved good results by optimizing the shifted log-variance. Given that this objective is only introduced in this work, maybe I would phrase this differently, since in previous work it was only incorporated as a heuristic for homogenizing the solution, rather than an optimization problem.
- How significant is the $O(d^3)$ computation cost associated with SDERF and ADERF? I guess the previous layer could easily have an overall dimension in the 1000s, where this could become a bottleneck on GPUs in terms of memory?
- In line 214, the authors make the remark that assuming $L \geq d$, the $O(Ld^2)$ cost dominates the $O(d^3)$ cost. Is this not already assumed in Theorem 4.2 (and hence, I assume in Theorem 4.3 from the way it is phrased), since this is necessary for the nonsingularity of $M^{(1)}$ and $M^{(2)}$?
- If the main improvement of SDERF compared to ADERF comes from allowing a non-isotropic $A$, would it be beneficial to define extensions of GERF or SADERF, which add this flexibility to the parametrization? What might be the computational cost of such an RF construction? This is just a hunch, but if this can be solved for without matrix decompositions, then we might have the best of both worlds? No problem if working this out is difficult, just curious.
- What is the range of sequence lengths in the speech modelling benchmark? Does this count as a long-range task?
- In the NLP task, what is the number of random features (number of MC samples) used? Is there any intuition, investigation that the authors could report about how to choose this hyperparameter?
- One more question; in the Appendix it is stated that replacing the standard normal distribution on $\omega$ with an orthogonal one works better in practice. is this achieved by QR decomposing a set of Gaussian vectors? Is there any intuition about why this improves the performance?

**Limitations:**

The paper is mainly theoretical, but it has direct implications regarding Transformers, which carry with themselves a variety of societal and environmental impacts, but this seems to be addressed.

---

> ### Author Rebuttal · Authors · 2023-08-08
>
> We would like to sincerely thank the Reviewer for all the comments and a favorable score. We address weaknesses and questions below.
>
> > I was missing some more comparisons between the variations: for one, SADERFs were not included in any of them, and only used for the last NLP task.
>
> Note that we report ADERFs in Figure 2 which can be thought as a stronger version of SADERFs. It doesn’t perform worse than the previous best variant (GERFs), however SDERFs is even better. For that reason, we evaluated SDERFs in the speech modelling experiment, however we weren’t able to use it in the NLP setup because we experienced unhealthy behaviour when using SVD and eigen decomposition in Jax. That’s why we chose SADERFs in this setup and in general we recommend using it when matrix decomposition is infeasible.
>
> > Another question that seems unaddressed to me is how much information is lost compared to a full Transformer on truly long-range tasks.
>
> We refer to [14] which is the original paper proposing RFs in the context of Transformers and demonstrating superior performance compared to full Transformers. We note that RFs proposed in our paper are a stronger variant of FAVOR+ from [14].
>
> > Is sigma analogous to the bandwidth parameter of the kernel in Figure 1 (right) and Figure 2?
>
> We don’t parametrize our Gaussian kernel definition (line 58) but sigma in Figures 1, 2 is equivalent to the inverse bandwidth parameter in the standard Gaussian kernel exp(-||x - y||^2 / (2 bandwidth^2)) since the arguments to (our) kernel are sigma x, sigma y.
>
> > Given that this objective is only introduced in this work, maybe I would phrase this differently…
>
> Thanks for the suggestion, we will incorporate it in the final version to mitigate a potential confusion. What we meant is that inherently [30] optimize Eq. (8) without knowing it and get good results which suggests that (8) is a good variance proxy.
>
> >  I guess the previous layer could easily have an overall dimension in the 1000s, where this could become a bottleneck on GPUs in terms of memory?
>
> Note that d is not the dimension of the previous layer but the dimension of the attention head which is much smaller and is typically 64.
>
> > Is this not already assumed in Theorem 4.2 (and hence, I assume in Theorem 4.3 from the way it is phrased) … ?
>
> That’s correct, thank you for your observation! We will add this clarification to the Theorem statements.
>
> > If the main improvement of SDERF compared to ADERF comes from allowing a non-isotropic A, would it be beneficial to define extensions of GERF or SADERF, which add this flexibility to the parametrization?
>
> Thank you for suggesting this, this could be a nice extension of our methods and we leave it to the future work.
>
> > What is the range of sequence lengths in the speech modelling benchmark?
>
> For speech modeling the max sequence length was ~900.
>
> > In the NLP task, what is the number of random features (number of MC samples) used? Is there any intuition, investigation that the authors could report about how to choose this hyperparameter?
>
> The number of RFs is M=128.  Bigger M can only decrease the variance of the approximation, hence we recommend setting M as higher as possible according to the given computational limitations. In our NLP experiments we tried 64, 128 and 256 and found that 128, 256 perform similar to each other while being a bit better to M=64. Moreover in Figure 3 we have shown plots for how accuracy changes across different dataset as we vary M.
>
> > One more question; in the Appendix it is stated that replacing the standard normal distribution on omega  with an orthogonal one works better in practice. Is this achieved by QR decomposing a set of Gaussian vectors? Is there any intuition about why this improves the performance?
>
> There is a thread of work about orthogonal random features and why they reduce variance for kernel estimation: see “Orthogonal Random Features”, Yu et al. 2016, [13, 14, 15] and their references/citations.

---

### Official Review · Reviewer_r6PB · 2023-07-26

**Soundness:** 3 good
**Presentation:** 2 fair
**Contribution:** 2 fair
**Rating:** 4
**Confidence:** 4

**Summary:**

This work focuses on positive linear features for softmax kernels which are relevant for accelerating kernel methods and transformers. The paper observes that the functional form for random Fourier features (GERFs) proposed by prior work in [30] can be seen as optimizing the shift-ed log-variance objective. Building on GERFs, a more expressive and parameterized functional form is proposed, and corresponding shifted-log variance is derived. After that, several simplifications are considered for improved optimization of the log-shifted variance of the random features, which leads to different forms of RFs that are easily computable while keeping subquadratic complexity in the number of sequence lengths. Towards the end, a few empirical results are presented in support of their contributions.

**Strengths:**

* This work studies an important class of problems that has broader applicability. The paper builds on observations/heuristics from the prior work and attempts to justify using theoretical arguments. During this exercise, the proposed methodologies are mathematically elegant and present several exciting results on RFs that are relevant to both kernel methods and transformers.
* While I have not examined every mathematical proof, the derivation appears generally accepted, insightful, and praiseworthy.


**Weaknesses:**

* To establish that GERF minimizes the objective, the supporting argument is the solution of 8 matches with the heuristic proposed by the [30]. If I have understood it correctly, more is needed as for an objective to be reasonable, its behavior beyond the maximizer/minimizer needs to be understood. Also, is there any relation between this objective and the actual variance of the GERF estimate? Can Jensen inequality be used? This is one of the main results of this paper, and it needs to be better motivated.
* Notations could be improved. E.g., use a consistent format for scalars and matrices. L is a scalar. Avoid unnecessary precision; e.g., the upper subscript in lines 157 and 158 could have been avoided and stated as part of the text. Note that this work is built on math and therefore requires extra effort on styling to make it accessible for a larger audience.
* Pages 5-6-7 are not used carefully and do not present the best aspects of this work. You are considering simple formulations and repeating more or less the same procedure for obtaining close form solution of the objective, which again needs to be justified better in the first place. Also, before going into these details, give an overview of possible choices on the parameters space and justification. Summarizing these results as a corollary instead of Theorem 4.3 and 4.4 might be helpful.
* In line 217: “operations for which implementation has not yet matured in popular deep learning libraries with GPU and TPU support.” What is the evidence for this claim? Are you suggesting that eigen decomposition can’t be computed efficiently on GPUs using torch and TensorFlow?


**Questions:**

* Is it possible to repeat the proof of Theorem 4.2 for a setting where the log of features comes from a non-linear function, potentially a neural network? The main idea is to extend from the quadratic functions of w and x, use a more general parametric form, and optimize a relaxation of the variance.  Note that to ensure the unbiasedness of the kernel, estimation can be imposed using constraints similar to the proposed work. How would you compare and contrast with the current approach? If not, why is it not possible?
* Why are time versus accuracy results not reported anywhere, assuming scalability is one of the main motivations? When will these methods accelerate, and when will they not? Theoretical complexity may not account for constants that stem from implementation.
* What aspects of the implementation may prohibit practitioners interested in the fast self-attention method from considering these proposed methods?
* Respond to weakness 1.

**Limitations:**

Please take a look at the weakness and questions.

---

> ### Author Rebuttal · Authors · 2023-08-08
>
> Thank you for all the comments and kind words in strengths! We address weaknesses and questions below.
>
> > If I have understood it correctly, more is needed as for an objective to be reasonable, its behavior beyond the maximizer/minimizer needs to be understood.
>
> An intuitive understanding of this objective is that, if we assume that variances over different pairs of points are tightly concentrated around one point, then the objective is the logarithm of that point. Jensen’s inequality can be used to lower-bound the variance, however not to upper bound it since log is a concave function. Note that our experimental evidence (Figures 1 and 2) suggests that the minimization of (8) indeed leads to variance minimization where we get up to e^10 times variance reduction.
>
> > Notations could be improved.
> > Pages 5-6-7 are not used carefully and do not present the best aspects of this work.
>
> We will incorporate notation and style suggestions by the Reviewer in the final revision. Thank you!
>
> > Are you suggesting that eigen decomposition can’t be computed efficiently on GPUs using torch and TensorFlow?
>
> We used the Jax codebase and we experienced errors and unhealthy behavior when using SVD and eigen decomposition in Jax.
>
> > Is it possible to repeat the proof of Theorem 4.2 for a setting where the log of features comes from a non-linear function, potentially a neural network?
>
> The quadratic nature of the problem is essential for our close form solutions, meaning that we don’t see how our proofs can be extended to arbitrary nonlinear functions. Perhaps approximate solutions are feasible, we leave that to future work since it’s outside of the scope of this paper.
>
> > Why are time versus accuracy results not reported anywhere, assuming scalability is one of the main motivations? When will these methods accelerate, and when will they not?
>
> Our methods’ complexity grows linearly with the sequence length as opposed to quadratic growth for the standard self-attention. Hence, we are guaranteed to get improvements for long sequences (of order 1000 and above). We emphasize that, apart from empirical contributions, our paper is mainly theoretical and provides a significant extension of random features for the Gaussian kernel with nontrivial close form solutions (Theorems 4.2 and 4.3). We ask the Reviewer to take that into account.
>
> > What aspects of the implementation may prohibit practitioners interested in the fast self-attention method from considering these proposed methods?
>
> For smaller sequence lengths of order d (d is usually 64 and is the dimension of the attention head) our methods won’t give efficiency improvements over standard self-attention since O(LMd) will be close to O(L^2 d).

---

> > ### Comment · Reviewer_r6PB · 2023-08-12
> > **Thanks for your response.**
> >
> > Thank you for your detailed response.
> >
> > In light of other reviews and rebuttal responses, I would like to reiterate that this work makes exciting contributions to theory and practice. However, I still believe that the theoretical claims lack rigorous justifications, and the empirical evidence does not support the practice-relevant results while having impressive asymptotic guarantees. It is interesting to note that the rationale for the appropriateness of the objective comes from experiments, while the overall validity of the proposed work relies on theoretical guarantees.
> >
> > One possible way to improve the results would be to conduct simple experiments on transformer inference and demonstrate that the results are crucial and could help practitioners. Thanks.

---

### Decision · Program_Chairs · 2023-09-21

**Decision:**

Accept (poster)

**Comment:**

The submission is about scaling up kernel methods using the random feature (RF) approach. Particularly, the authors concentrate on the problem of efficient multiplication/approximation with the Gram matrix associated to the softmax kernel (or more generally to the scaled softmax kernel defined in line 55), which is a commonly-appearing operation in various kernel algorithms. They propose the dense-exponential random feature class (DERF, line 145-146) extending the A,B,C scalar parameters of the family GERF (generalized exponential RF) family to matrices. After showing that these features are RFs for the softmax kernel (meant in the sense of Def. 2.1) and providing analytical expression for their variance (Theorem 4.1), they specialize the construction to ADERF (asymmetric dense exponential RFs), SDERF (symmetric dense-exponential RFs) and SADERF (simplified ADERF) where one can get the variance-minimizing parameters in closed form (Theorem 4.2, 4.3). The approach is illustrated in kernel regression and self-attention approximation.

Providing principled techniques to scale up kernel methods is an important problem in contemporary data science; the considered task is of clear interest to the NeurIPS community. The presented approach is novel, with relevance from both empirical and theoretical point of view, as it was elaborated by the reviewers.

Note: line 22: random features for operator-valued kernels: The authors might wish to cite "Romain Brault, Markus Heinonen, Florence d'Alché-Buc. Random Fourier Features For Operator-Valued Kernels, Asian Conference on Machine Learning (ACML), pages 110-125, 2016" as the first work in the area.